# Misspecified Gaussian Process Bandit Optimization

**Ilija Bogunovic**
ETH Zürich

**Andreas Krause**
ETH Zürich

## Abstract

We consider the problem of optimizing a black-box function based on noisy bandit feedback. Kernelized bandit algorithms have shown strong empirical and theoretical performance for this problem. They heavily rely on the assumption that the model is well-specified, however, and can fail without it. Instead, we introduce a *misspecified* kernelized bandit setting where the unknown function can be $\epsilon$–uniformly approximated by a function with a bounded norm in some Reproducing Kernel Hilbert Space (RKHS). We design efficient and practical algorithms whose performance degrades minimally in the presence of model misspecification. Specifically, we present two algorithms based on Gaussian process (GP) methods: an optimistic EC-GP-UCB algorithm that requires knowing the misspecification error, and Phased GP Uncertainty Sampling, an elimination-type algorithm that can adapt to unknown model misspecification. We provide upper bounds on their cumulative regret in terms of $\epsilon$, the time horizon, and the underlying kernel, and we show that our algorithm achieves optimal dependence on $\epsilon$ with no prior knowledge of misspecification. In addition, in a stochastic contextual setting, we show that EC-GP-UCB can be effectively combined with the regret bound balancing strategy and attain similar regret bounds despite not knowing $\epsilon$.

## 1   Introduction

Bandit optimization has been successfully used in a great number of machine learning and real-world applications, e.g., in mobile health [42], environmental monitoring [40], economics [27], hyperparameter tuning [26], to name a few. To scale to large or continuous domains, modern bandit approaches try to model and exploit the problem structure that is often manifested as correlations in rewards of "similar" actions. Hence, the key idea of *kernelized bandits* is to consider only smooth reward functions of a low norm belonging to a chosen Reproducing Kernel Hilbert Space (RKHS) of functions. This permits the application of flexible nonparametric Gaussian process (GP) models and Bayesian optimization methods via a well-studied link between RKHS functions and GPs (see, e.g., [18] for a concise review).

A vast majority of previous works on nonparametric kernelized bandits have focused on designing algorithms and theoretical bounds on the standard notions of regret (see, e.g., [40, 9, 35]). However, they solely focus on the *realizable* (i.e., well-specified) case in which one assumes perfect knowledge of the true function class. For example, the analysis of the prominent GP-UCB [40] algorithm assumes the model to be well-specified and ignores potential misspecification issues. As the realizability assumption may be too restrictive in real applications, we focus on the case where it may only hold *approximately*. In practice, model misspecifications can arise due to various reasons, such as incorrect choice of kernel, consideration of an overly smooth function class, hyperparameter estimation errors, etc. Hence, an open question is to characterize the impact of model misspecification in the kernelized setting, and to design robust algorithms whose performance degrades optimally with the increasing level of misspecification.

35th Conference on Neural Information Processing Systems (NeurIPS 2021).

In this paper, we study the GP bandit problem with model misspecification in which the true unknown function might be $\epsilon$-far (as measured in the max norm) from a member of the learner's assumed hypothesis class. We propose a novel GP bandit algorithm and regret bounds that depend on the misspecification error, time horizon, and underlying kernel. Specifically, we present an algorithm that is based on the classical *uncertainty sampling* approach that is frequently used in Bayesian optimization and experimental design. Importantly, our main presented algorithm assumes *no knowledge* of the misspecification error $\epsilon$ and achieves standard regret rates in the realizable case.

**Related work on GP bandits.** GP bandit algorithms have received significant attention in recent years (e.g., [40, 11, 9, 39]). While the most popular approaches in the stochastic setting rely on *upper confidence bound (UCB)* and *Thompson sampling* strategies, a number of works also consider *uncertainty sampling* procedures (e.g., [10, 41, 6]). Beyond the standard setting, numerous works have also considered the *contextual* bandit setting (e.g., [22, 44, 21]), while the case of unknown kernel hyperparameters and misspecified smoothness has been studied in, e.g., [46, 3, 47]. [46, 3] assume that the reward function is smooth as measured by the known (or partially known) kernel, while in our problem, we allow the unknown function to lie outside a given kernel's reproducing space.

Related corruption-robust GP bandits in which an adversary can additively perturb the observed rewards are recently considered in [4]. In Section 2, we also discuss how the misspecified problem can be seen from this perspective, where the corruption function is fixed and the adversarial budget scales with the time horizon. While the focus of [4] is on protecting against an adaptive adversary and thus designing randomized algorithms and regret bounds that depend on the adversarial budget, we propose deterministic algorithms and analyze the impact of misspecification.

Apart from corruptions, several works have considered other robust aspects in GP bandits, such as designing robust strategies against the shift in uncontrollable covariates [5, 20, 31, 38, 7]. While they report robust regret guarantees, they still assume the realizable case only. Our goal of attaining small regret despite the wrong hypothesis class requires very different techniques from these previous works.

**Related work on misspecified linear bandits.** Recently, works on reinforcement learning with misspecified linear features (e.g., [12, 45, 17]) have renewed interest in the related misspecified *linear* bandits (e.g., [48], [24], [30], [15]) first introduced in [16]. In [16], the authors show that standard algorithms must suffer $\Omega(\epsilon T)$ regret under an additive $\epsilon$–perturbation of the linear model. Recently, [48] propose a robust variant of OFUL [1] that *requires knowing* the misspecification parameter $\epsilon$. In particular, their algorithm obtains a high-probability $\tilde{O}(d\sqrt{T} + \epsilon\sqrt{d}T)$ regret bound. In [24], the authors propose another arm-elimination algorithm based on G-experimental design. Unlike the previous works, their algorithm is *agnostic* to the misspecification level, and its performance matches the lower bounds. As shown in [24], when the number of actions $K$ is large ($K \gg d$), the "price" of $\epsilon$-misspecification must grow as $\Omega(\epsilon\sqrt{d}T)$.[1] Our main algorithm is inspired by the proposed technique for the finite-arm misspecified linear bandit setting [24]. It works in the more general kernelized setting, uses a simpler data acquisition rule often used in Bayesian optimization and experimental design, and recovers the same optimal guarantees when instantiated with linear kernel.

Several works have recently considered the misspecified *contextual* linear bandit problem with *unknown* model misspecification $\epsilon$. [14] introduce a new family of algorithms that require access to an online oracle for square loss regression and address the case of adversarial contexts. Concurrent work of [33] solves the case when contexts / action sets are stochastic. Both works ([14] and [33]) leverage CORRAL-type aggregation [2] of contextual bandit algorithms and achieve the optimal $\tilde{O}(\sqrt{d}T\epsilon + d\sqrt{T})$ regret bound. Finally, in [32], the authors present a practical *master* algorithm that plays *base* algorithms that come with a candidate regret bound that may not hold during all rounds. The master algorithm plays base algorithms in a *balanced* way and suitably eliminates algorithms whose regret bound is no longer valid. Similarly to the previous works that rely on the CORRAL-type master algorithms, we use the balancing master algorithm of [33] together with our GP-bandit base algorithm to provide contextual misspecified regret bounds.

Around the time of the submission of this work, a related approach that among others also considers misspecified kernel bandits appeared online [8]. [8, Theorem 3] contains the same regret scaling due to misspecification as we obtain in our results. The main difference between the two works is in the proposed algorithms and analysis techniques. Our approach does not require robust estimators and simply uses the standard ones (i.e., GP posterior/ kernelized ridge regression mean and variance

---

[1] A result by [15] further shows that this result can be improved to $O(\epsilon\sqrt{K}T)$ in the small-$K$ regime.

estimators) that can be computed in the closed-form. It can also handle infinite action sets similarly to the classical Bayesian optimization algorithms; it utilizes a single acquisition function that, in practice, can be maximized via standard off-the-shelf global optimization solvers. We also present a complete treatment of the misspecified problem by showing the failure of standard UCB approaches, algorithms for known and unknown $\epsilon$, an impossibility result, and extensions to the contextual bandit setting. Finally, our main algorithm (Algorithm 2) demonstrates the use of a different acquisition function in comparison to [24, 8] that relies on standard and non-robust estimators.

**Contributions.** In this paper, we systematically handle model misspecification in GP bandits. Specifically, this paper makes the following contributions:

- We introduce a misspecified kernelized bandit problem, and for known misspecification error $\epsilon$, we present the EC-GP-UCB algorithm with enlarged confidence bounds that achieves cumulative $R_T = O(B\sqrt{\gamma_T T} + \gamma_T\sqrt{T} + \epsilon T\sqrt{\gamma_T})$ regret. Our simple lower bound argument shows that $\Omega(T\epsilon)$ regret is unavoidable in the general kernelized setting.
- For when $\epsilon$ is unknown, we propose another algorithm based on uncertainty sampling and phased exploration that achieves (up to $\mathrm{polylog}$ factors) the previous regret rates in the misspecified setting, and standard regret guarantees in the realizable case (when $\epsilon = 0$).
- Finally, we consider a misspecified *contextual* kernelized problem, and show that when action sets are stochastic, our EC-GP-UCB algorithm can be effectively combined with the regret bound balancing strategy from [33] to achieve previous regret bounds (up to some additive lower order terms).

## 2   Problem statement

We consider the problem of sequentially maximizing some black-box reward function $f^* : \mathcal{D} \to \mathbb{R}$ over a known compact set of actions $\mathcal{D} \subset \mathbb{R}^d$. To learn about $f^*$, the learner relies on sequential noisy bandit feedback, i.e., at every round $t$, the learner selects $x_t \in \mathcal{D}$ and obtains a noisy observation

$$y_t^* = f^*(x_t) + \eta_t, \tag{1}$$

where we assume independent (over rounds) $\sigma$-sub-Gaussian noise (see Appendix A for definition).

Let $k(\cdot, \cdot)$ denote a positive definite kernel function defined on $\mathcal{D} \times \mathcal{D}$ and $\mathcal{H}_k(\mathcal{D})$ be its associated Reproducing Kernel Hilbert Space (RKHS) of well-behaved functions. Suppose that before making any decision, the learner is provided with a hypothesis class

$$\mathcal{F}_k(\mathcal{D}; B) = \{ f \in \mathcal{H}_k(\mathcal{D}) : \|f\|_k \le B \}, \tag{2}$$

where every member function has a bounded RKHS norm $\|f\|_k = \sqrt{\langle f, f \rangle}$ for some $B > 0$, that measures the complexity of $f$ with respect to kernel $k(\cdot, \cdot)$. We consider kernel functions such that $k(x, x) \le 1$ for every $x \in \mathcal{D}$. Most commonly used kernel functions that satisfy this property are outlined in Appendix A.

The standard setting assumes a *realizable* (i.e., well-specified) scenario in which $f^* \in \mathcal{F}_k(\mathcal{D}; B)$, i.e., the unknown function is a member of the known RKHS with bounded norm. In contrast, in the *misspecified* setting, we assume that the learner is informed that $f^*$ can be uniformly approximated by a member from the given hypothesis class $\mathcal{F}_k(\mathcal{D}; B)$, i.e.,

$$\min_{f \in \mathcal{F}_k(\mathcal{D};B)} \|f - f^*\|_\infty \le \epsilon, \tag{3}$$

for some $\epsilon > 0$ and max-norm $\|\cdot\|_\infty$. Here, we note that if two functions are close in RKHS norm, then they are also pointwise close but the reverse does not need to be true. Hence, the true function $f^*$ can in principle have a significantly larger RKHS norm than any $f$ from $\mathcal{F}_k(\mathcal{D}; B)$, or it might not even belong to the considered RKHS.

We also note that in the case of continuous *universal* kernels [28] (i.e., due to the universal function approximation property of such kernels), any continuous function $f^*$ on $\mathcal{D}$ satisfies the above assumption (Eq. (3)) for any $\epsilon > 0$ and $\mathcal{F}_k(\mathcal{D}; B)$ with suitably large RKHS norm bound $B$. This is a difference in comparison to the previously studied misspecified linear setting, and another motivation to study the misspecified kernelized problem.

As in the standard setting (which corresponds to the case when $\epsilon = 0$), we assume that $k(\cdot, \cdot)$ and $B$ are known to the learner. The goal of the learner is to *minimize* the cumulative regret

$$R_T^* = \sum_{t=1}^{T} \left( \max_{x \in \mathcal{D}} f^*(x) - f^*(x_t) \right), \tag{4}$$

where $T$ is a time horizon. We denote instantaneous regret at round $t$ as $r_t^* = f^*(x^*) - f^*(x_t)$, where $x^* \in \arg\max_{x \in \mathcal{D}} f^*(x)$, and note that $R_T^* = \sum_{t=1}^{T} r_t^*$. The learner's algorithm knows $\mathcal{F}_k(\mathcal{D}; B)$, meaning that it takes $\mathcal{D}$, $k(\cdot, \cdot)$ and $B$ as input. Crucially, in our most general considered problem variant, the exact value of misspecification $\epsilon$ is assumed to be *unknown*, and so we seek an algorithm that can adapt to any $\epsilon > 0$, including the realizable case when $\epsilon = 0$.

Alternatively, one can consider competing with a best-in-class benchmark, i.e.,

$$R_T = \sum_{t=1}^{T} \left( \max_{x \in \mathcal{D}} \tilde{f}(x) - \tilde{f}(x_t) \right), \tag{5}$$

where

$$\tilde{f} \in \arg\min_{f \in \mathcal{F}_k(\mathcal{D}; B)} \|f - f^*\|_\infty \tag{6}$$

and $|\tilde{f}(x) - f^*(x)| \le \epsilon$ for every $x \in \mathcal{D}$. Here, the goal is to minimize cumulative regret in case the true unknown objective is $\tilde{f} \in \mathcal{F}_k(\mathcal{D}; B)$, while noisy observations of $\tilde{f}$ that the learner receives are at most $\epsilon$-misspecified, i.e.,

$$y_t^* = \underbrace{\tilde{f}(x_t) + m(x_t)}_{f^*(x_t)} + \eta_t, \tag{7}$$

where $m(\cdot) : \mathcal{D} \to [-\epsilon, \epsilon]$ is a fixed and unknown function. Since it holds that $\|\tilde{f} - f^*\|_\infty \le \epsilon$, the absolute difference between $R_T$ and $R_T^*$ is at most $2\epsilon T$, and as will already become clear in Section 3.2, this difference will have no significant impact on the scalings in our main bound (since $R_T^* = \Omega(\epsilon T)$). In our theoretical analysis, we will interchangeably use both regret definitions (from Eqs. (4) and (5)).

## 3   Algorithms for misspecified kernelized bandits

We start this section by recalling a Gaussian Process (GP) framework for learning RKHS functions. Then, we present different GP-bandit algorithms and theoretical regret bounds for misspecified settings of increasing levels of difficulty.

### 3.1   Learning with Gaussian Processes

In the realizable case (i.e., when the true unknown $f \in \mathcal{F}_k(\mathcal{D}; B)$ and the learner knows the true hypothesis space $\mathcal{F}_k(\mathcal{D}; B)$) and under the noise model described in Eq. (1), uncertainty modeling and learning in standard GP-bandit algorithms can be viewed through the lens of Gaussian Process models. A Gaussian Process $GP(\mu(\cdot), k(\cdot, \cdot))$ over the input domain $\mathcal{D}$, is a collection of random variables $(f(x))_{x \in \mathcal{D}}$ where every finite number of them $(f(x_i))_{i=1}^{n}$, $n \in \mathbb{N}$, is jointly Gaussian with mean $\mathbb{E}[f(x_i)] = \mu(x_i)$ and covariance $\mathbb{E}[(f(x_i) - \mu(x_i))(f(x_j) - \mu(x_j))] = k(x_i, x_j)$ for every $1 \le i, j \le n$. Standard algorithms implicitly use a zero-mean $GP(0, k(\cdot, \cdot))$ as the prior distribution over $f$, i.e., $f \sim GP(0, k(\cdot, \cdot))$, and assume that the noise variables are drawn independently across $t$ from $\mathcal{N}(0, \lambda)$ with $\lambda > 0$. After collecting new data, that is, a sequence of actions $\{x_1, \ldots, x_t\}$ and their corresponding noisy observations $\{y_1, \ldots, y_t\}$, the posterior distribution under previous assumptions is also Gaussian with the mean and variance that can be computed in closed-form as:

$$\mu_t(x) = k_t(x)^T (K_t + \lambda I_t)^{-1} Y_t \tag{8}$$

$$\sigma_t^2(x) = k(x, x) - k_t(x)^T (K_t + \lambda I_t)^{-1} k_t(x), \tag{9}$$

where $k_t(x) = [k(x_1, x), \ldots, k(x_t, x)]^T \in \mathbb{R}^{t \times 1}$, $K_t = [k(x_s, x_{s'})]_{s, s' \ge t} \in \mathbb{R}^{t \times t}$ is the corresponding kernel matrix, and $Y_t := [y_1, \ldots, y_t]$ denotes a vector of observations.

The previous standard modeling assumptions lead itself to model misspecifications as GP samples are rougher than RKHS functions and are not contained in $\mathcal{H}_k(\mathcal{D})$ with high probability. Although this

leads to a mismatched hypothesis space, GPs and RKHS functions are closely related (see, e.g., [18]) when used with same kernel function, and it is possible to use GP models to infer reliable confidence intervals on the unknown $f \in \mathcal{F}_k(\mathcal{D}; B)$. Under this assumption, the popular algorithms such as GP-UCB [40] construct statistical confidence bounds that contain $f$ with high probability uniformly over time horizon, i.e., the following holds $|f(x) - \mu_{t-1}(x)| \leq \beta_t \sigma_{t-1}(x)$ for every $t \geq 1$ and $x \in \mathcal{D}$. Here, $\{\beta_t\}_{t \leq T}$ stands for the sequence of parameters that are suitably set (see Lemma 1) to (i) trade-off between exploration and exploitation and (ii) ensure the validity of the confidence bounds. In every round $t$, GP-UCB then queries the unknown function at a point $x_t \in \mathcal{D}$ that maximizes the upper confidence bound given by $\mu_{t-1}(\cdot) + \beta_t \sigma_{t-1}(\cdot)$, with $\mu_{t-1}(\cdot)$ and $\sigma_{t-1}(\cdot)$ as defined in Eqs. (8) and (9). In the standard setting, GP-UCB is *no regret*, meaning that $R_T/T \to 0$ as $T \to \infty$. However, in the misspecified setting, the previous standard confidence bounds are no longer valid and one needs to consider different strategies.

Before moving to the misspecified case, we recall an important kernel-dependent quantity known as *maximum information gain* $\gamma_t(k, \mathcal{D})$ [40] that is frequently used to characterize the regret bounds.[2] It stands for the maximum amount of information that a set of noisy observations can reveal about the unknown function sampled from a zero-mean Gaussian process with kernel $k$. Specifically, for a set of points $S \subset \mathcal{D}$, we use $f_S$ to denote a random vector $[f(x)]_{x \in S}$, and $Y_S$ to denote the corresponding noisy observations obtained as $Y_S = f_S + \eta_S$, where $\eta_S \sim \mathcal{N}(0, \lambda I)$. The maximum information gain is then defined as:

$$\gamma_t(k, \mathcal{D}) := \max_{S \subset \mathcal{D}:|S|=t} I(f_S, Y_S) = \max_{S \subset \mathcal{D}:|S|=t} \frac{1}{2}|I_t + \lambda^{-1} K_t|, \tag{10}$$

where $I(\cdot, \cdot)$ stands for the mutual information between random variables, and $|\cdot|$ is the determinant. Simply put, if samples are taken "close" to each other (far from each other) as measured by the kernel, they are more correlated (less correlated) under the GP prior and provide less (more) information. As shown in [40], the maximum information gain $\gamma_t(k, \mathcal{D})$ scales sublinearly with $t$ for the most commonly used kernels (see Appendix A).

### 3.2 Known misspecification error and optimistic approaches

In this section, we also use the "well-specified" mean $\mu_t(\cdot)$ and variance $\sigma_t^2(\cdot)$ estimates from Eqs. (8) and (9), where we assume that noisy observations used in Eq. (8) correspond to a function $\tilde{f} \in \arg\min_{f \in \mathcal{F}_k(\mathcal{D};B)} \|f - f^*\|_\infty$. We note that $\sigma_t^2(\cdot)$ does not depend on the observations (i.e., $Y_t$), and additionally, we define $\mu_t^*(\cdot)$ that depends on the noisy observations of the true $f^*$, i.e.,

$$\mu_t^*(x) = k_t(x)^T (K_t + \lambda I_t)^{-1} Y_t^*, \tag{11}$$

where $Y_t^* := [y_1^*, \ldots, y_t^*]$, and $y_i^* = f^*(x_t) + \eta_t$ for $1 \leq i \leq t$. The only difference between the definitions of $\mu_t^*(x)$ and $\mu_t(x)$ comes from the used observation vector, i.e., $Y_t^*$ and $Y_t$, respectively.

We also use the following standard result from [40, 9, 13] that provides confidence bounds around the unknown function in the realizable setting.

**Lemma 1.** *Let $f(\cdot)$ be a function that belongs to the space of functions $\mathcal{F}_k(\mathcal{D}; B)$. Assume the $\sigma$-sub-Gaussian noise model as in Eq. (1), and let $Y_{t-1} := [y_1, \ldots, y_{t-1}]$ denote the vector of previous noisy observations that correspond to the queried points $(x_1, \ldots, x_{t-1})$. Then, the following holds with probability at least $1 - \delta$ simultaneously over all $t \geq 1$ and $x \in \mathcal{D}$:*

$$|f(x) - \mu_{t-1}(x)| \leq \beta_t \sigma_{t-1}(x), \tag{12}$$

*where $\mu_{t-1}(\cdot)$ and $\sigma_{t-1}(\cdot)$ are given in Eq. (8) and Eq. (9) with $\lambda > 0$, and*

$$\beta_t = \frac{\sigma}{\lambda^{1/2}} \Big(2\ln(1/\delta) + \sum_{t'=1}^{t-1} \ln(1 + \lambda^{-1}\sigma_{t'-1}(x_{t'}))\Big)^{\frac{1}{2}} + B. \tag{13}$$

Next, we start by addressing the misspecified setting when $\epsilon$ is *known* to the learner. We consider minimizing $R_T$ (from Eq. (5)), and provide an upper confidence bound algorithm with enlarged confidence bounds EC-GP-UCB (see also Algorithm 1):

$$x_t \in \arg\max_{x \in \mathcal{D}} \mu_{t-1}^*(x) + \Big(\beta_t + \frac{\epsilon\sqrt{t}}{\sqrt{\lambda}}\Big)\sigma_{t-1}(x), \tag{15}$$

---

[2]We often use notation $\gamma_t$ in the text when $k(\cdot, \cdot)$ and $\mathcal{D}$ are clear from context.

---

**Algorithm 1** EC-GP-UCB (Enlarged Confidence GP-UCB)

---
1: **Require:** Kernel function $k(\cdot, \cdot)$, domain $\mathcal{D}$, misspecification $\epsilon$, and parameters $B, \lambda, \sigma$
2: Set $\mu_0(x) = 0$ and $\sigma_0(x) = k(x, x)$, for all $x \in \mathcal{D}$
3: **for** $t = 1, \ldots, T$ **do**
4:     Choose

$$x_t \in \arg\max_{x \in \mathcal{D}} \mu_{t-1}^*(x) + \left(\beta_t + \frac{\epsilon\sqrt{t}}{\sqrt{\lambda}}\right)\sigma_{t-1}(x) \tag{14}$$

5:     Observe $y_t^* = f^*(x_t) + \eta_t$
6:     Update to $\mu_t^*(\cdot)$ and $\sigma_t(\cdot)$ by using $(x_t, y_t)$ according to Eq. (9) and Eq. (11)
7: **end for**

---

where the confidence interval enlargement is to account for the use of the biased mean estimator $\mu_{t-1}^*(\cdot)$ (instead of $\mu_{t-1}(\cdot)$). This can be interpreted as introducing an additional exploration bonus to the standard GP-UCB algorithm [40] in case of misspecification. The enlargement corresponds to the difference in the mean estimators that is captured in the following lemma:

**Lemma 2.** *For any $x \in \mathcal{D}$, $t \geq 1$ and $\lambda > 0$, we have*

$$|\mu_t(x) - \mu_t^*(x)| \leq \frac{\epsilon\sqrt{t}}{\sqrt{\lambda}}\sigma_t(x), \tag{16}$$

*where $\mu_t(\cdot)$ and $\mu_t^*(\cdot)$ are defined as in Eq. (8) and Eq. (11), respectively, and $\sigma_t(\cdot)$ is from Eq. (9).*

Next, we upper bound the cumulative regret of the proposed algorithm.

**Theorem 1.** *Suppose the learner's hypothesis class is $\mathcal{F}_k(\mathcal{D}; B)$ for some fixed $B > 0$ and $\mathcal{D} \subset \mathbb{R}^d$. For any $f^*$ defined on $\mathcal{D}$ and $\epsilon \geq 0$ such that $\min_{f \in \mathcal{F}_k(\mathcal{D};B)} \|f - f^*\|_\infty \leq \epsilon$, EC-GP-UCB with enlarged confidence Eq. (15) and known $\epsilon$, achieves the following regret bound with probability at least $1 - \delta$:*

$$R_T = O\left(B\sqrt{\gamma_T T} + \sqrt{(\ln(1/\delta) + \gamma_T)\gamma_T T} + \epsilon T\sqrt{\gamma_T}\right). \tag{17}$$

As remarked before, the regret bound from Eq. (17) implies the upper bound on $R_T^*$ of the same order, since $R_T^*$ and $R_T$ differ by at most $2\epsilon T$. The first part of the bound, i.e., $O\big(B\sqrt{\gamma_T T} + \sqrt{(\ln(1/\delta) + \gamma_T)\gamma_T T}\big)$, corresponds to the standard regret bound achieved by GP-UCB in the realizable scenario [40, 9] which is also known to nearly match the lower bounds in case of commonly used kernels [36]. On the other hand, in Appendix C, we also demonstrate that $\epsilon T$ dependence is unavoidable in general for any algorithm in the misspecified kernelized setting.

A similar robust GP-UCB algorithm with enlarged confidence bounds has been first considered in [4] to defend against adversarial corruptions. One can think of the misspecified setting as a corrupted one where, at every round, the corruption is bounded by $\epsilon$, yielding a total corruption budget of $C = T\epsilon$. The algorithm proposed in [4] attains a $C\sqrt{\gamma_T T}$ regret bound (due to corruptions) which is strictly suboptimal in comparison to the $\epsilon T\sqrt{\gamma_T}$ bound obtained in our Theorem 1.

Finally, we note that to obtain the previous result, the algorithm requires knowledge of $\epsilon$ as input, and it is unclear how to adapt the algorithm to the unknown $\epsilon$ case. In particular, we show in Appendix B.2 that the problem in the analysis arises since there is no effective way of controlling the uncertainty at $x^*$ when using standard UCB-based approaches. To address this more practical setting, in the next section we propose our main algorithm that does *not* require the knowledge of $\epsilon$.

### 3.3 Unknown misspecification error: Phased GP Uncertainty Sampling

Our second proposed Phased GP Uncertainty Sampling algorithm that has no knowledge of the true $\epsilon$ parameter is shown in Algorithm 2. It runs in episodes of exponentially increasing length $m_e$ and maintains a set of potentially optimal actions $\mathcal{D}_e$. In each episode $e$, actions are selected via exploration-encouraging uncertainty sampling, i.e., an action of maximal GP epistemic uncertainty $\sigma_t(\cdot)$ is selected (with ties broken arbitrarily). The selected action is then used to update $\sigma_t(\cdot)$, that does not depend on the received observations (see Eq. (9)). We also note that at the beginning of every episode, the algorithm reverts back to the prior model (before any data is observed), i.e., by setting $\mu_0(x) = 0$ and $\sigma_0(x) = k(x, x)$, for all $x \in \mathcal{D}_e$.

---

**Algorithm 2** Phased GP Uncertainty Sampling

---

1: **Require:** Kernel function $k(\cdot,\cdot)$, domain $\mathcal{D}$, and parameters $B$, $\lambda$, $\sigma$
2: Set episode index $e = 1$, episode length $m_e = 1$, and set of potentially optimal actions $\mathcal{D}_e = \mathcal{D}$
3: Set $\mu_0(x) = 0$ and $\sigma_0(x) = k(x,x)$, for all $x \in \mathcal{D}_e$
4: **for** $t = 1, \ldots, m_e$ **do**
5:     Choose

$$x_t \in \arg\max_{x \in D_e} \sigma_{t-1}^2(x) \tag{19}$$

6:     Update to $\sigma_t(\cdot)$ by including $x_t$ according to Eq. (9)
7: **end for**
8: Receive $\{y_1, \ldots, y_{m_e}\}$, such that

$$y_t = f^*(x_t) + \eta_t \quad \text{for} \quad t \in \{1, \ldots, m_e\},$$

    and use them to compute $\mu_{m_e}^*(\cdot)$ according to Eq. (11)
9: Set

$$\mathcal{D}_{e+1} \leftarrow \big\{ x \in \mathcal{D}_e : \mu_{m_e}^*(x) + \beta_{m_e+1}\sigma_{m_e}(x) \geq \max_{x \in \mathcal{D}_e} \big( \mu_{m_e}^*(x) - \beta_{m_e+1}\sigma_{m_e}(x) \big) \big\}, \tag{20}$$

10: $m_{e+1} \leftarrow 2m_e, e \leftarrow e + 1$ and return to step (3) (terminate after $T$ total function evaluations)

---

After every episode, the algorithm receives $m_e$ noisy observations which are then used to update the mean estimate $\mu_{m_e}^*(\cdot)$ according to Eq. (8). Finally, the algorithm *eliminates* actions that appear suboptimal according to the current but potentially wrong model. This is done by retaining all the actions $x \in \mathcal{D}_e$ whose *hallucinated* upper confidence bound is at least as large as the maximum hallucinated lower bound, that is,

$$\mu_{m_e}^*(x) + \beta_{m_e+1}\sigma_{m_e}(x) \geq \max_{x \in \mathcal{D}_e} \big( \mu_{m_e}^*(x) - \beta_{m_e+1}\sigma_{m_e}(x) \big), \tag{18}$$

where $\beta_{m_e+1}$ is set as in the realizable setting. Here, the term "hallucinated" refers to the fact that these confidence bounds might be invalid (for $\tilde{f}$), and hence, the optimal action might be eliminated. We also note that in case $\epsilon = 0$, the hallucinated confidence bounds are actually valid. In our analysis (see Appendix D), we prove that although the optimal action can be eliminated, a "near-optimal" one is retained after every episode. Hence, we only need to characterize the difference in regret due to such possible wrong elimination within all episodes (whose number is logarithmic in $T$).

Our Algorithm 2 bears some similarities with the Phased Elimination algorithm of [24] designed for the related *linear* misspecified bandit setting. Both algorithms employ an episodic action elimination strategy. However, the algorithm of [24] crucially relies on the Kiefer-Wolfowitz theorem [19] and requires computing a near-optimal design at every episode (i.e., a probability distribution over a set of currently plausibly optimal actions) that minimizes the worst-case variance of the resulting least-squares estimator. Adapting this approach to the kernelized setting and RKHS function classes of infinite dimension is a nontrivial task. Instead, we use a kernel ridge regression estimate and we show that it is sufficient to sequentially select actions via a simple acquisition rule that does not rely on finite-dimensional feature approximations of the kernel. In particular, our algorithm, at each round $t$ (in episode $e$), selects a *single* action $x$ (from $\mathcal{D}_e$) that maximizes $\|\phi(x)\|^2_{(\Phi_t^*\Phi_t + \lambda I_k)^{-1}}$ (here, $\phi(x)$ denotes kernel features, i.e., $k(x,x') = \langle \phi(x), \phi(x') \rangle_k$), which is equivalent to maximizing the GP posterior variance in Eq. (19) (see Appendix D for details). We also note that similar GP uncertainty sampling acquisition rules are commonly used in Bayesian optimization and experimental design (see, e.g., [10, 6, 23]), and efficient iterative updates of the posterior variance that avoid computation of a $t \times t$ kernel matrix inverse at each round are also available (see, e.g., Appendix F in [9]).

In the next theorem, we bound the cumulative regret of the proposed algorithm and use $\tilde{O}(\cdot)$ notation to hide $\text{polylog}(T)$ factors.

**Theorem 2.** *Suppose the learner's hypothesis class is $\mathcal{F}_k(\mathcal{D}; B)$ for some fixed $B > 0$ and $\mathcal{D} \subset \mathbb{R}^d$. For any $f^*$ defined on $\mathcal{D}$ and $\epsilon \geq 0$ such that $\min_{f \in \mathcal{F}_k(\mathcal{D};B)} \|f - f^*\|_\infty \leq \epsilon$, Phased GP Uncertainty Sampling (Algorithm 2) achieves the following regret bound with probability at least $1 - \delta$:*

$$R_T^* = \tilde{O}\Big( B\sqrt{\gamma_T T} + \sqrt{(\ln(1/\delta) + \gamma_T)\gamma_T T} + \epsilon T\sqrt{\gamma_T} \Big).$$

In comparison with the previous kernel-based EC-GP-UCB algorithm, our phased uncertainty sampling algorithm attains the same regret guarantee without knowledge of $\epsilon$. Our result holds in the case of infinite action sets, and further on, we can substitute the existing upper bounds on $\gamma_T(k, \mathcal{D})$ to specialize it to particular kernels. For example, in the case of linear kernel and compact domain, we have $\gamma_T(k_{\text{lin}}, \mathcal{D}) = O(d \log T)$, while for squared-exponential kernel it holds $\gamma_t(k_{\text{SE}}, \mathcal{D}) = O((\log T)^{d+1})$ [40]. Moreover, when the used kernel is linear, we recover the same misspecification regret rate of [24], i.e., $\tilde{O}(\epsilon T \sqrt{d})$.

## 4 Algorithm for the *contextual* misspecified kernelized bandit setting

In this section, we consider a contextual misspecified problem with unknown $f^* : \mathcal{D} \to [0, 1]$ and the same assumptions as before (see Section 2). The main difference comes from the fact that at every round $t$, the learner needs to choose an action from a possibly different action set $\mathcal{D}_t \subseteq \mathcal{D}$. We assume that the learner observes a context $c_t \triangleq \{x\}_{x \in D_t}$ at every round $t$, where $c_t$ is assumed to be drawn i.i.d. from some distribution.[3] The learner then plays $x_t \in \mathcal{D}_t$ and observes $y_t = f^*(x_t) + \eta_t$, with 1-sub-Gaussian noise and independence between time steps.

Since we consider the setting in which the set of actions changes with every round, we note that algorithms that employ action-elimination strategies, such as our Phased GP Uncertainty Sampling, are not easily adapted to this scenario as they require a fixed action set. On the other hand, our optimistic EC-GP-UCB algorithm requires knowing the true misspecification parameter $\epsilon$ a priori, which is also unsatisfactory. To address this, in this section, we combine our EC-GP-UCB with the *regret bound balancing* strategy from [32].

We measure the regret incurred by the learner by using the corresponding contextual regret definition:

$$R_T^* = \sum_{t=1}^{T} \big( \max_{x \in \mathcal{D}_t} f^*(x) - f^*(x_t) \big). \tag{21}$$

We specialize the regret bound balancing strategy to the GP-bandit setting. The proposed algorithmic solution considers $M$ different *base* algorithms $(\text{Base})_{i=1}^M$, and uses an elimination $M$–armed bandit *master* algorithm (see Algorithm 3) to choose which base algorithm to play at every round.

**Base algorithms.** At each round, the learner plays according to the suggestion of a single *base* algorithm $i \in \{1, \ldots, M\}$. We use $\tau_i(t)$ to denote the set of rounds in which base algorithm $i$ is selected up to time $t$, and $N_i(t) = |\tau_i(t)|$ to denote the number of rounds algorithm $i$ is played by the learner. After a total of $T$ rounds, the regret of algorithm $i$ is given by $R_{i,T}^* = \sum_{t \in \tau_i(T)} \big( \max_{x \in \mathcal{D}_t} f^*(x) - f^*(x_t) \big)$, and we also note that $R_T^* = \sum_{i=1}^M R_{i,T}^*$.

---
**Algorithm 3** Regret bound balancing [33]
---
**Require:** $(\text{Base})_{i=1}^M$
**Initialize:** $I_1 = [M]$, $N_i(0) = 0$ for all $i$
**for** $t = 1, \ldots, T$ **do**
    Receive $\mathcal{D}_t$
    Select $i_t \in \arg\min_{i \in I_t} \mathcal{R}^{(i)}(N_i(t-1))$
    $\text{Base}_{i_t}$ plays $x_t \in \mathcal{D}_t$ and observes $y_t$
    Update $\text{Base}_{i_t}$ and set $N_i(t) = N_i(t) + 1$
    Update the set of active algorithms $I_{t+1}$
**end for**
---

As a good candidate for base algorithms, we use our EC-GP-UCB algorithm (from Algorithm 1), but instead of the true unknown $\epsilon$, we instantiate EC-GP-UCB with different candidate values $\hat{\epsilon}_i$. In particular, we use $M = \lceil 1 + \frac{1}{2} \log_2(T/\gamma_T^2) \rceil$ different EC-GP-UCB algorithms, and for every algorithm $i \in [M]$, we set $\hat{\epsilon}_i = \frac{2^{1-i}}{\sqrt{\gamma_T}}$. This means that as $i$ decreases, the used algorithms are more robust and can tolerate larger misspecification error. On the other hand, overly conservative values lead to excessive exploration and large regret. We consider the true $\epsilon$ to be in $[0, 1)$, and our goal is to ideally match the performance of EC-GP-UCB that is run with such $\epsilon$.

---

[3]Our approach can be extended to the general setting where $f(x, c)$ is defined on the joint action-context space, and where contexts are assumed to be drawn i.i.d. (e.g., user profiles are often considered as contexts in recommendation systems [25], and i.i.d. is a reasonable assumption for different users).

Each base algorithm comes with a *candidate* regret upper bound $\mathcal{R}_i(t) : \mathbb{N} \to \mathbb{R}_+$ (that holds with high probability) of the form given in Theorem 1. Moreover, every candidate regret bound $\mathcal{R}_i(t)$ is non-decreasing in $t$ and $\mathcal{R}_i(0) = 0$. Without loss of generality, we can assume that candidate bounds can increase by at most 1 per each round, i.e., $0 \leq \mathcal{R}_i(t) - \mathcal{R}_i(t-1) \leq 1$ for all rounds $t$ and algorithms $i$. We use the following forms of EC-GP-UCB candidate bounds in Algorithm 3:

$$\mathcal{R}_i(N_i(t)) = \min\{c_1' \gamma_T \sqrt{N_i(t)} + c_2' \hat{\epsilon}_i \sqrt{\gamma_T} N_i(t), \; N_i(t)\}, \tag{22}$$

for some quantities $c_1', c_2'$ that do not depend on $N_i(t)$ or $\hat{\epsilon}_i$. For a given $\hat{\epsilon}_i$, these bounds are computable for every $t \in [T]$. In particular, to evaluate them, we also need to compute $\gamma_T$ (Eq. (10)). We note that for popularly used kernels, analytical upper bounds exist as discussed in Section 3.3, while in general, we can efficiently approximate it up to a constant factor via simple greedy maximization.[4]

We refer to algorithm $i$ as being *consistent* if $R_{i,t}^* \leq \mathcal{R}_i(N_i(t))$ for all $t \in [T]$ with high probability, and otherwise as *inconsistent*. We also let $i^*$ denote the first consistent algorithm $i$ for which $\hat{\epsilon}_{i^*} \geq \epsilon$. In particular, we have that $\hat{\epsilon}_i \leq \hat{\epsilon}_{i^*}$ for every inconsistent algorithm $i$.

**Master algorithm.** We briefly recall the regret bound balancing algorithm of [32] and its main steps. At each round, the learner selects an algorithm $i_t$ according to the regret bound balancing principle:

$$i_t \in \underset{i \in I_t}{\arg\min} \, \mathcal{R}_i(N_i(t-1)), \tag{23}$$

where $I_t \subseteq [M]$ is the set of *active* (i.e., plausibly consistent) algorithms at round $t$. Then, the learner uses the algorithm $i_t$ to select $x_t$, observes reward $y_t$, and increases $N_i(t) = N_i(t-1) + 1$. Intuitively, by evaluating candidate regret bounds for different number of times, the algorithm keeps them almost equal and hence balanced, i.e., $\mathcal{R}_i(N_i(t)) \leq \mathcal{R}_j(N_j(t)) + 1$ for every $t$ and all active algorithms $i, j \in I_t$.

The remaining task is to update the set of active algorithms based on the received observation. We define $J_i(t) = \sum_{j \in \tau_i(t)} y_j$ as the cumulative reward of algorithm $i$ in the first $t$ rounds. As in [32] (Section 4), we set $I_1 = [M]$, and at the end of each round, the learner eliminates every inconsistent algorithm $i \in I_t$ from $I_{t+1}$ when:

$$J_i(t) + \mathcal{R}_i(N_i(t)) + c\sqrt{N_i(t)\ln(M\ln N_i(t)/\delta)} < \max_{j \in I_t} J_j(t) - c\sqrt{N_j(t)\ln(M\ln N_j(t)/\delta)}, \tag{24}$$

where $c$ is a suitably set absolute constant (see Lemma A.1 in [32]). Crucially, the elimination condition from Eq. (24) is used in Algorithm 3, and it ensures that no consistent algorithm is eliminated (with high probability). In particular, since each $\mathcal{D}_t$ is sampled i.i.d., it holds that $N_i(t)\mathbb{E}_{\mathcal{D}_t}[\max_{x \in \mathcal{D}_t} f^*(x)]$ is upper/lower bounded by the left/right hand side of Eq. (24). Hence, if the upper bound of algorithm $i$ for this quantity is smaller than the maximum lower bound, the algorithm can be classified as inconsistent and consequently eliminated from $I_{t+1}$.

**Regret bound.** By using the general regret result for Algorithm 3 from [32] (see Appendix E), we show the total regret obtained for Algorithm 3 when instantiated with our EC-GP-UCB algorithm and candidate regret bounds provided in Eq. (22). This is formally captured in the following proposition.

**Proposition 1.** *Consider $M = \lceil 1 + \frac{1}{2}\log_2(T/\gamma_T^2) \rceil$ EC-GP-UCB base algorithms, each with candidate regret upper bound of the form provided in Eq. (22) with $\hat{\epsilon}_i = \frac{2^{1-i}}{\sqrt{\gamma_T}}$ for each $i \in [M]$. In the misspecified kernelized contextual setting, when run with such $M$ algorithms, Algorithm 3 achieves the following regret bound with probability at least $1 - M\delta$:*

$$R_T^* = \tilde{O}\Big((B\sqrt{\gamma_T} + \gamma_T)\sqrt{T} + \epsilon T\sqrt{\gamma_T} + (B\sqrt{\gamma_T} + \gamma_T)^2\Big). \tag{25}$$

The obtained contextual regret bound recovers (up to poly-logarithmic factors and an additive term of lower-order) the bound of the EC-GP-UCB instance (from Theorem 1) that assumes the knowledge of the true $\epsilon$.

---

[4]When applied to Eq. (10), a solution found by a simple and efficient greedy algorithm achieves at least $(1 - 1/e)^{-1}\gamma_T$ (this is due to submodularity of the mutual information; see, e.g., [40]).

## 5   Conclusion

We have considered the GP-bandit optimization problem in the case of a misspecified hypothesis class. Our work systematically handles model misspecifications in GP bandits and provides robust and practical algorithms. We designed novel algorithms based on ideas such as enlarged confidence bounds, phased epistemic uncertainty sampling, and regret bound balancing, for the standard (with known/unknown misspecification error) and contextual settings. While there have been some theoretical [29] and practical [37] efforts to quantify the impact of misspecified priors in the related Bayesian setting [40], an interesting direction for future work is to develop algorithms and regret bounds in case of misspecified GP priors.

## Acknowledgments and Disclosure of Funding

This project has received funding from the European Research Council (ERC) under the European Unions Horizon 2020 research and innovation programme grant agreement No 815943 and ETH Zürich Postdoctoral Fellowship 19-2 FEL-47.

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
