# Supplementary Material

## Misspecified GP Bandit Optimization

### Ilija Bogunovic and Andreas Krause (NeurIPS 2021)

## A  GP bandits: Useful definitions and auxiliary results (Realizable setting)

**Assumed observation model.** We say a real-valued random variable $X$ is $\sigma$-sub-Gaussian if it its mean is zero and for all $\varepsilon \in \mathbb{R}$ we have

$$\mathbb{E}[\exp(\varepsilon X)] \leq \exp\left(\frac{\sigma^2 \varepsilon^2}{2}\right). \tag{26}$$

At every round $t$, the learner selects $x_t \in D$ and observes the noisy function evaluation

$$y_t = f(x_t) + \eta_t, \tag{27}$$

where we assume $\{\eta_t\}_{t=1}^T$ are $\sigma$-sub-Gaussian random variables that are independent over time steps. Such assumptions on the noise variables are frequently used in bandit optimization.

Typically, in kernelized bandits, we assume that unknown $f \in \mathcal{F}_k(\mathcal{D}; B) = \{f \in \mathcal{H}_k(\mathcal{D}) : \|f\|_k \leq B\}$, where $\mathcal{H}_k(\mathcal{D})$ is the reproducing kernel Hilbert space of functions associated with the given positive-definite kernel function. Typically, the learner knows $\mathcal{F}_k(\mathcal{D}; B)$, meaning that both $k(\cdot, \cdot)$ and $B$ are considered as input to the learner's algorithm.

**Example kernel functions.** We outline some commonly used kernel functions $k : \mathcal{D} \times \mathcal{D} \to \mathbb{R}$, that we also consider:

- Linear kernel: $k_{\text{lin}}(x, x') = x^T x'$,

- Squared exponential kernel: $k_{\text{SE}}(x, x') = \exp\left(-\frac{\|x - x'\|^2}{2l^2}\right)$,

- Matérn kernel: $k_{\text{Mat}}(x, x') = \frac{2^{1-\nu}}{\Gamma(\nu)}\left(\frac{\sqrt{2\nu}\|x-x'\|}{l}\right) J_\nu\left(\frac{\sqrt{2\nu}\|x-x'\|}{l}\right)$,

where $l$ denotes the length-scale hyperparameter, $\nu > 0$ is an additional hyperparameter that dictates the smoothness, and $J(\nu)$ and $\Gamma(\nu)$ denote the modified Bessel function and the Gamma function, respectively [34].

**Maximum information gain.** Maximum information gain is a *kernel-dependent* quantity that measures the complexity of the given function class. It has first been introduced in [40], and since then it has been used in numerous works on Gaussian process bandits. Typically, the upper regret bounds in Gaussian process bandits are expressed in terms of this complexity measure.

It represents the maximum amount of information that a set of noisy observations can reveal about the unknown $f$ that is sampled from a zero-mean Gaussian process with kernel $k$, i.e., $f \sim GP(0, k)$. More precisely, for a set of sampling points $S \subset \mathcal{D}$, we use $f_S$ to denote a random vector $[f(x)]_{x \in S}$, and $Y_S$ to denote the corresponding noisy observations obtained as $Y_S = f_S + \eta_S$, where $\eta_S \sim \mathcal{N}(0, \lambda I)$. We note that under this setup after observing $Y_S$, the posterior distribution of $f$ is a Gaussian process with posterior mean and variance that correspond to Eq. (8) and Eq. (9).

The maximum information gain (about $f$) after observing $t$ noisy samples is defined as (see [40]):

$$\gamma_t(k, \mathcal{D}) := \max_{S \subset \mathcal{D}:|S|=t} I(f_S; Y_S) = \max_{S \subset \mathcal{D}:|S|=t} \frac{1}{2}|I_t + \lambda^{-1}K_t|, \tag{28}$$

where $I(\cdot, \cdot)$ denotes the mutual information between random variables, $|\cdot|$ is used to denote a matrix determinant, and $K_t$ is a kernel matrix $[k(x_s, x_{s'})]_{s,s' \leq t} \in \mathbb{R}^{t \times t}$.

Under the previous setup (GP prior and Gaussian likelihood), the maximum information gain can be expressed in terms of predictive GP variances:

$$\gamma_t(k, \mathcal{D}) = \max_{\{x_1, \ldots, x_t\} \subset \mathcal{D}} \frac{1}{2} \sum_{s=1}^t \ln\left(1 + \lambda^{-1}\sigma_{s-1}^2(x_s)\right). \tag{29}$$

The proof of this claim can be found in [40, Lemma 5.3]. It also allows us to rewrite Eq. (12) from Lemma 1 in the following frequently used form:

$$|f(x) - \mu_{t-1}(x)| \leq \left( \frac{\sigma}{\lambda^{1/2}} \sqrt{2\ln(1/\delta) + 2\gamma_{t-1}} + B \right) \sigma_{t-1}(x). \tag{30}$$

Next, we outline an important relation (due to [40]) frequently used to relate the sum of GP predictive standard deviations with the maximum information gain. We use the formulation that follows from Lemma 4 in [9]:

**Lemma 3.** *Consider some kernel $k : \mathcal{D} \times \mathcal{D} \to \mathbb{R}$ such that $k(x,x) \leq 1$ for every $x \in \mathcal{D}$, and let $f \sim GP(0, k)$ be a sample from a zero-mean GP with the corresponding kernel function. Then for any set of queried points $\{x_1, \ldots, x_t\}$ and $\lambda > 0$, it holds that*

$$\sum_{i=1}^{t} \sigma_{i-1}(x_i) \leq \sqrt{(2\lambda + 1)\gamma_t t}. \tag{31}$$

Finally, we outline bounds on $\gamma_t(k, \mathcal{D})$ for commonly used kernels as provided in [40]. An important observation is that the maximum information gain is sublinear in terms of number of samples $t$ for these kernels.

**Lemma 4.** *Let $d \in \mathbb{N}$ and $\mathcal{D} \subset \mathbb{R}^d$ be a compact and convex set. Consider a kernel $k : \mathcal{D} \times \mathcal{D} \to \mathbb{R}$ such that $k(x,x) \leq 1$ for every $x \in \mathcal{D}$, and let $f \sim GP(0, k)$ be a sample from a zero-mean Gaussian Process (supported on $\mathcal{D}$) with the corresponding kernel function. Then in case of*

- *Linear kernel: $\gamma_t(k_{lin}, \mathcal{D}) = O(d \log t)$,*

- *Squared exponential kernel: $\gamma_t(k_{SE}, \mathcal{D}) = O((\log t)^{d+1})$,*

- *Matérn kernel: $\gamma_t(k_{Mat}, \mathcal{D}) = O(t^{d(d+1)/(2\nu+d(d+1))} \log t)$.*

We also note that the previous rates in case of the Matérn kernel have been recently improved to: $O\left( t^{\frac{d}{2\nu+d}} (\log t)^{\frac{2\nu}{2\nu+d}} \right)$ in [43].

# B Proofs from Section 3.2 (EC-GP-UCB)

## B.1 EC-GP-UCB with known misspecification

**Theorem 1.** *Suppose the learner's hypothesis class is $\mathcal{F}_k(\mathcal{D}; B)$ for some fixed $B > 0$ and $\mathcal{D} \subset \mathbb{R}^d$. For any $f^*$ defined on $\mathcal{D}$ and $\epsilon \geq 0$ such that $\min_{f \in \mathcal{F}_k(\mathcal{D};B)} \|f - f^*\|_\infty \leq \epsilon$, EC-GP-UCB with enlarged confidence Eq. (15) and known $\epsilon$, achieves the following regret bound with probability at least $1 - \delta$:*

$$R_T = O\Big(B\sqrt{\gamma_T T} + \sqrt{(\ln(1/\delta) + \gamma_T)\gamma_T T} + \epsilon T \sqrt{\gamma_T}\Big). \tag{17}$$

*Proof.* From the definition of $R_T$ in Eq. (4) (also recall the definition of $\tilde{f}$ from Eq. (6)), we have:

$$R_T = \sum_{t=1}^{T} \big(\max_{x \in \mathcal{D}} \tilde{f}(x) - \tilde{f}(x_t)\big) \tag{32}$$

$$\leq \sum_{t=1}^{T} \max_{x \in \mathcal{D}} \Big(\mu_{t-1}^*(x) + \big(\beta_t + \tfrac{\epsilon\sqrt{t}}{\sqrt{\lambda}}\big)\sigma_{t-1}(x)\Big) - \Big(\mu_{t-1}^*(x_t) - \big(\beta_t + \tfrac{\epsilon\sqrt{t}}{\sqrt{\lambda}}\big)\sigma_{t-1}(x_t)\Big) \tag{33}$$

$$\leq \sum_{t=1}^{T} \mu_{t-1}^*(x_t) + \big(\beta_t + \tfrac{\epsilon\sqrt{t}}{\sqrt{\lambda}}\big)\sigma_{t-1}(x_t) - \mu_{t-1}^*(x_t) + \big(\beta_t + \tfrac{\epsilon\sqrt{t}}{\sqrt{\lambda}}\big)\sigma_{t-1}(x_t) \tag{34}$$

$$= \sum_{t=1}^{T} 2\big(\beta_t + \tfrac{\epsilon\sqrt{t}}{\sqrt{\lambda}}\big)\sigma_{t-1}(x_t) \tag{35}$$

$$= \sum_{t=1}^{T} 2\beta_t \sigma_{t-1}(x_t) + \sum_{t=1}^{T} 2\tfrac{\epsilon\sqrt{t}}{\sqrt{\lambda}}\sigma_{t-1}(x_t) \tag{36}$$

$$\leq 2\beta_T \sum_{t=1}^{T} \sigma_{t-1}(x_t) + 2\tfrac{\epsilon\sqrt{T}}{\sqrt{\lambda}} \sum_{t=1}^{T} \sigma_{t-1}(x_t) \tag{37}$$

$$\leq 2\beta_T \sqrt{(2\lambda + 1)\gamma_T T} + 2\tfrac{\epsilon\sqrt{T}}{\sqrt{\lambda}} \sqrt{(2\lambda + 1)\gamma_T T} \tag{38}$$

$$= 2\Big(\tfrac{\sigma}{\lambda^{1/2}}\sqrt{2\ln(1/\delta) + 2\gamma_T} + B\Big)\sqrt{(2\lambda + 1)\gamma_T T} + 2\tfrac{\epsilon}{\sqrt{\lambda}}T\sqrt{(2\lambda + 1)\gamma_T} \tag{39}$$

$$= O\Big(B\sqrt{\gamma_T T} + \sqrt{(\ln(1/\delta) + \gamma_T)\gamma_T T} + \epsilon T \sqrt{\gamma_T}\Big). \tag{40}$$

where Eq. (33) follows from the validity of the enlarged confidence bounds (by combining Lemmas 1 and 2) and Eq. (34) follows from the selection rule of EC-GP-UCB (Eq. (14)). Finally, Eq. (38) is due to Eq. (31), and Eq. (39) follows by upper-bounding $\beta_T$ as in Eq. (30). $\square$

## B.2 EC-GP-UCB and unknown misspecification

In this section, we outline the main hindrance with the analysis of EC-GP-UCB (or GP-UCB [40]) when $\epsilon$ is unknown. We start with the definition of $R_T$ (Eq. (4)) and repeat the initial steps as in Eq. (33):

$$R_T = \sum_{t=1}^{T} \big(\max_{x \in \mathcal{D}} \tilde{f}(x) - \tilde{f}(x_t)\big) \tag{41}$$

$$\leq \sum_{t=1}^{T} \mu_{t-1}^*(x^*) + \big(\beta_t + \tfrac{\epsilon\sqrt{t}}{\sqrt{\lambda}}\big)\sigma_{t-1}(x^*) - \Big(\mu_{t-1}^*(x_t) - \big(\beta_t + \tfrac{\epsilon\sqrt{t}}{\sqrt{\lambda}}\big)\sigma_{t-1}(x_t)\Big). \tag{42}$$

Since $\epsilon$ is unknown here, we cannot repeat the analysis from the previous section as the learner cannot choose:

$$\arg\max_{x \in \mathcal{D}} \mu_{t-1}^*(x) + \big(\beta_t + \tfrac{\epsilon\sqrt{t}}{\sqrt{\lambda}}\big)\sigma_{t-1}(x).$$

Instead, it can select:

$$x_t \in \arg\max_{x \in \mathcal{D}} \mu_{t-1}^*(x) + \beta_t \sigma_{t-1}(x),$$

which corresponds to the standard GP-UCB algorithm when $\epsilon = 0$. By using this rule in Eq. (42), we can arrive at:

$$R_T \leq \sum_{t=1}^{T} \mu_{t-1}^*(x^*) + \beta_t \sigma_{t-1}(x^*) + \tfrac{\epsilon\sqrt{t}}{\sqrt{\lambda}}\sigma_{t-1}(x^*) - \left(\mu_{t-1}^*(x_t) - \left(\beta_t + \tfrac{\epsilon\sqrt{t}}{\sqrt{\lambda}}\right)\sigma_{t-1}(x_t)\right) \quad (43)$$

$$\leq \sum_{t=1}^{T} \mu_{t-1}^*(x_t) + \beta_t \sigma_{t-1}(x_t) + \tfrac{\epsilon\sqrt{t}}{\sqrt{\lambda}}\sigma_{t-1}(x^*) - \left(\mu_{t-1}^*(x_t) - \left(\beta_t + \tfrac{\epsilon\sqrt{t}}{\sqrt{\lambda}}\right)\sigma_{t-1}(x_t)\right) \quad (44)$$

$$= \sum_{t=1}^{T} 2\beta_t \sigma_{t-1}(x_t) + \tfrac{\epsilon\sqrt{t}}{\sqrt{\lambda}}\sigma_{t-1}(x_t) + \tfrac{\epsilon\sqrt{t}}{\sqrt{\lambda}}\sigma_{t-1}(x^*). \quad (45)$$

While the first two terms in this bound can be effectively controlled and bounded as in the proof of Theorem 1, the last term, i.e., $\sum_{t=1}^{T} \tfrac{\epsilon\sqrt{t}}{\sqrt{\lambda}}\sigma_{t-1}(x^*)$, poses an issue since we cannot ensure that $\sum_{t=1}^{T} \sigma_{t-1}(x^*)$ is decaying with $t$, similarly to $\sum_{t=1}^{T} \sigma_{t-1}(x_t)$.

## C   Optimal dependence on misspecification parameter

In this section, we argue that a joint dependence on $T\epsilon$ is unavoidable in cumulative regret bounds. We consider the noiseless case, and we let the domain be the unit hypercube $\mathcal{D} = [0,1]^d$. Consider some function $f(x)$ defined on $\mathcal{D}$ such that $f(x) \in [-2\zeta, 2\zeta]$ for every $x \in D$. Moreover, let $f(x)$ satisfy the constant RKHS norm bound $B$, and let there exist a non-empty region $\mathcal{W} \subset \mathcal{D}$ where $f(x) \geq \zeta$ for every $x \in \mathcal{W}$. Such a function can easily be constructed, e.g., via the approach outlined in [36].

Now suppose that $\epsilon = 2\zeta$ and let the true unknown function $f^*$ be 0 everywhere in $\mathcal{D}$, except at a single point $x \in \mathcal{W}$ where it is $2\zeta$. Hence, any algorithm that tries to optimize $f^*$ will only observe 0-values almost surely, since sampling at the point where the function value is $\epsilon = 2\zeta$ is a zero-probability event. Hence, after $T$ rounds, regardless of the sampling algorithm, $\Omega(\epsilon T)$ regret will be incurred. Finally, it is not hard to see that $\|f - f^*\|_\infty \leq 2\zeta = \epsilon$, and so there exists a function of bounded RKHS norm that is $\epsilon$ pointwise close to $f^*$.

## D   Proofs from Section 3.3 (Phased GP Uncertainty Sampling)

We start this section by outlining the following auxiliary lemma and then we proceed with the proof of Theorem 2. The following lemma provides an upper bound on the difference between the mean estimators obtained from querying the true and best-in-class functions, respectively. Here, for the sake of analysis, we use $\mu_t(\cdot)$ to denote the hypothetical mean estimator in case $m$ noisy observations of the best-in-class function are available.

**Lemma 2.** *For any $x \in \mathcal{D}$, $t \geq 1$ and $\lambda > 0$, we have*

$$|\mu_t(x) - \mu_t^*(x)| \leq \tfrac{\epsilon\sqrt{t}}{\sqrt{\lambda}}\sigma_t(x), \quad (16)$$

*where $\mu_t(\cdot)$ and $\mu_t^*(\cdot)$ are defined as in Eq. (8) and Eq. (11), respectively, and $\sigma_t(\cdot)$ is from Eq. (9).*

*Proof.* Our proof closely follows the one of [4, Lemma 2], with the problem-specific difference at the very end of the proof (see Eq. (53)).

Let $x$ be any point in $\mathcal{D}$, and fix a time index $t \geq 1$. Recall that $Y_t^* = [y_1^*, \ldots, y_t^*]$ where each $y_i^*$ for $i \leq t$, is obtained as in Eq. (1). Following upon Eq. (7), we can write $\tilde{Y}_t = [y_1^* - m(x_1), \ldots, y_t^* - m(x_t)]$ which correspond to the hypothetical noisy observations of the function belonging to the learner's RKHS. From the definitions of $\mu_t(\cdot)$ and $\mu_t^*(\cdot)$, we have:

$$|\mu_t^*(x) - \mu_t(x)| = |k_t(x)^T(K_t + \lambda I_t)^{-1}Y_t^* - k_t(x)^T(K_t + \lambda I_t)^{-1}\tilde{Y}_t| \quad (46)$$

$$= |k_t(x)^T(K_t + \lambda I_t)^{-1}m_t|, \quad (47)$$

where $m_t = [m(x_1), \ldots, m(x_t)]$. We proceed by upper bounding the absolute difference, i.e., $|k_t(x)^T(K_t + \lambda I_t)^{-1}m_t|$, but first we define some additional terms.

Let $\mathcal{H}_k(\mathcal{D})$ denote the learner's hypothesis space, i.e., RKHS of functions equipped with inner-product $\langle \cdot, \cdot \rangle_k$ and corresponding norm $\| \cdot \|_k$. This space is completely determined by its associated $k(\cdot, \cdot)$ that satisfies: (i) $k(x, \cdot) \in \mathcal{H}_k(\mathcal{D})$ for all $x \in \mathcal{D}$ and (ii) $f(x) = \langle f, k(x, \cdot) \rangle_k$ for all $x \in \mathcal{D}$ (reproducing property). Due to these two properties and by denoting $\phi(x) := k(x, \cdot)$, we can write $k(x, x') = \langle k(x, \cdot), k(x', \cdot) \rangle_k = \langle \phi(x), \phi(x') \rangle_k$ for all $x, x' \in D$. Moreover, let $\Phi_t$ denote operator $\Phi_t : \mathcal{H}_k(\mathcal{D}) \to \mathbb{R}^t$, such that for every $f \in \mathcal{H}_k(\mathcal{D})$ and $i \in \{1, \ldots, t\}$, we have $(\Phi_t f)_i = \langle \phi(x_i), f \rangle_k$, and also let $\Phi_t^*$ denote its adjoint $\Phi_t^* : \mathbb{R}^t \to \mathcal{H}_k(\mathcal{D})$. We can then write $K_t = \Phi_t \Phi_t^*$, and $k_t(x) = \Phi_t \phi(x)$. We also define the weighted norm of vector $x$, by $\|x\|_\Phi = \sqrt{\langle x, \Phi x \rangle}$.

By using the following property of linear operators:

$$(\Phi_t^* \Phi_t + \lambda I_k)^{-1} \Phi_t^* = \Phi_t^* (\Phi_t \Phi_t^* + \lambda I_t)^{-1},$$

we first have:

$$|k_t(x)^T (K_t + \lambda I_t)^{-1} m_t| = |\langle (\Phi_t^* \Phi_t + \lambda I_k)^{-1} \phi(x), \Phi_t^* m_t \rangle_k| \tag{48}$$

$$\leq \|(\Phi_t^* \Phi_t + \lambda I_k)^{-1/2} \phi(x)\|_k \|(\Phi_t^* \Phi_t + \lambda I_k)^{-1/2} \Phi_t^* m_t\|_k \tag{49}$$

$$= \|\phi(x)\|_{(\Phi_t^* \Phi_t + \lambda I_k)^{-1}} \|\Phi_t^* m_t\|_{(\Phi_t^* \Phi_t + \lambda I_k)^{-1}} \tag{50}$$

$$= \lambda^{-1/2} \sigma_t(x) \sqrt{\langle \Phi_t \Phi_t^* m_t, (\Phi_t \Phi_t^* + \lambda I_t)^{-1} m_t \rangle} \tag{51}$$

$$= \lambda^{-1/2} \sigma_t(x) \sqrt{m_t^T K_t (K_t + \lambda I_t)^{-1} m_t} \tag{52}$$

$$\leq \lambda^{-1/2} \sigma_t(x) \sqrt{\lambda_{\max} \left( K_t (K_t + \lambda I_t)^{-1} \right) \|m_t\|_2^2} \tag{53}$$

$$\leq \frac{\epsilon \sqrt{t}}{\lambda^{1/2}} \sigma_t(x), \tag{54}$$

where Eq. (49) is by Cauchy-Schwartz, and Eq. (51) follows from the following standard identity (see, e.g., Eq. (50) in [4]):

$$\sigma_t(x) = \lambda^{1/2} \|\phi(x)\|_{(\Phi_t^* \Phi_t + \lambda I_k)^{-1}}. \tag{55}$$

Finally, $\lambda_{\max}(\cdot)$ denotes a maximum eigenvalue in Eq. (53), and Eq. (54) follows since for $\lambda > 0$, we have $\lambda_{\max}(K_t (K_t + \lambda I_t)^{-1}) \leq 1$, as well as by upper bounding $\|m_t\|_2 \leq \sqrt{t} \|m_t\|_\infty$ where $\|m_t\|_\infty \leq \epsilon$ which holds by definition of $m(\cdot)$ (see Eq. (7)). $\qquad \square$

Now, we are ready to state the proof of Theorem 2.

**Theorem 2.** *Suppose the learner's hypothesis class is $\mathcal{F}_k(\mathcal{D}; B)$ for some fixed $B > 0$ and $\mathcal{D} \subset \mathbb{R}^d$. For any $f^*$ defined on $\mathcal{D}$ and $\epsilon \geq 0$ such that $\min_{f \in \mathcal{F}_k(\mathcal{D}; B)} \|f - f^*\|_\infty \leq \epsilon$, Phased GP Uncertainty Sampling (Algorithm 2) achieves the following regret bound with probability at least $1 - \delta$:*

$$R_T^* = \tilde{O}\left( B \sqrt{\gamma_T T} + \sqrt{(\ln(1/\delta) + \gamma_T) \gamma_T T} + \epsilon T \sqrt{\gamma_T} \right).$$

*Proof.* We present the proof by splitting it into three main parts. We start with episodic misspecification.

**Episodic misspecification.** First, we bound the absolute difference between the misspecified mean estimator $\mu_{m_e}^*(\cdot)$ (from Eq. (11)) and *best-in-class* function $f \in \mathcal{F}_k(D; B)$ (i.e., $f \in \arg\min_{f \in \mathcal{F}_k(\mathcal{D}; B)} \|f - f^*\|_\infty \leq \epsilon$) at the end of some arbitrary episode $e$. Also, $\mu_m(\cdot)$ is defined in Eq. (8), where noisy observations in the definition correspond to $f(\cdot)$.

For any $x \in \mathcal{D}_e$, we have:

$$|\mu_{m_e}^*(x) - f(x)| = |\mu_{m_e}^*(x) + \mu_{m_e}(x) - \mu_{m_e}(x) - f(x)| \tag{56}$$

$$\leq |\mu_{m_e}^*(x) - \mu_{m_e}(x)| + |\mu_{m_e}(x) - f(x)| \tag{57}$$

$$\leq \frac{\epsilon \sigma_{m_e}(x) \sqrt{m_e}}{\sqrt{\lambda}} + \beta_{m_e+1} \sigma_{m_e}(x). \tag{58}$$

Here, Eq. (57) follows from triangle inequality, and Eq. (58) follows from Lemmas 1 and 2.

Next, we make the following observation:

$$\max_{x \in D} \sigma_{m_e}(x) \le \frac{1}{m_e} \sum_{t=1}^{m_e} \sigma_{t-1}(x_t) \tag{59}$$

$$\le \frac{1}{m_e} \sqrt{m_e (1 + 2\lambda) \gamma_{m_e}(k, \mathcal{D})} = \sqrt{\frac{(1 + 2\lambda) \gamma_{m_e}(k, \mathcal{D})}{m_e}}, \tag{60}$$

where Eq. (59) follows from the definition of $x_t$ (see Eq. (19)) and the fact that $\sigma_{t-1}(\cdot)$ is non-increasing in $t$. Finally, we used the result from Lemma 3 to arrive at Eq. (60) together with $\gamma_{m_e}(k, \mathcal{D}_e) \le \gamma_{m_e}(k, \mathcal{D})$ since $\mathcal{D}_e \subseteq \mathcal{D}$.

By upper bounding $\sigma_{m_e}(x)$ in the first term in Eq. (58) with $\max_{x \in D} \sigma_{m_e}(x)$, and by using the upper bound obtained in Eq. (60), we have that for any $x \in \mathcal{D}_e$, it holds:

$$|\mu_{m_e}^*(x) - f(x)| \le \beta_{m_e+1} \sigma_{m_e}(x) + \epsilon \sqrt{(2 + \lambda^{-1}) \gamma_{m_e}(k, \mathcal{D})}. \tag{61}$$

**Elimination.** Because the algorithm does not use the "valid" confidence bounds for the best-in-class $f$ in Eq. (20), it can eliminate its maximum even after the first episode. However, we show that there always remains a point $\hat{x}_e$ that is "close" to the best point (defined below) in every episode $e$.

Let $D_e$ denote the remaining points at the beginning of the episode $e$ (in Algorithm 2) and let $\hat{x}_e = \arg\max_{x \in \mathcal{D}_e} \{\mu_{m_e}^*(x) - \beta_{m_e+1} \sigma_{m_e}(x)\}$. We note that this point will remain in $\mathcal{D}_{e+1}$ according to the condition in Eq. (20) for retaining points. Next, we assume the worst-case scenario that the optimal point $x_e^* = \arg\max_{x \in \mathcal{D}_e} f(x)$ is eliminated at the end of the episode, i.e., $x_e^* \notin \mathcal{D}_{e+1}$. Then it holds, due to the condition Eq. (20) inside the algorithm that (note that both $\hat{x}_e$ and $x_e^*$ are in $\mathcal{D}_e$):

$$\mu_{m_e}^*(\hat{x}_e) - \beta_{m_e+1} \sigma_{m_e}(\hat{x}_e) > \mu_{m_e}^*(x_e^*) + \beta_{m_e+1} \sigma_{m_e}(x_e^*). \tag{62}$$

Next, by applying Eq. (61) to both sides, we obtain

$$f(\hat{x}_e) + \epsilon \sqrt{(2 + \lambda^{-1}) \gamma_{m_e}(\mathcal{D}_e)} > f(x_e^*) - \epsilon \sqrt{(2 + \lambda^{-1}) \gamma_{m_e}(\mathcal{D})}, \tag{63}$$

and by rearranging we obtain

$$2\epsilon \sqrt{(2 + \lambda^{-1}) \gamma_{m_e}(\mathcal{D})} > f(x_e^*) - f(\hat{x}_e), \tag{64}$$

which bounds the difference between the function values of the optimal (possibly eliminated from $\mathcal{D}_{e+1}$) point and the one that is retained. We also note that for $x_{e+1}^* = \arg\max_{x \in \mathcal{D}_{e+1}} f(x)$, it holds

$$2\epsilon \sqrt{(2 + \lambda^{-1}) \gamma_{m_e}(\mathcal{D})} > f(x_e^*) - f(x_{e+1}^*), \tag{65}$$

since $f(\hat{x}_e) \le f(x_{e+1}^*)$ and both $\hat{x}_e, x_{e+1}^* \in \mathcal{D}_{e+1}$.

**Regret.** We can now proceed to obtain the main regret bound. First, we show the following bound that holds for every $x \in D_e$. We let $x_e^* = \arg\max_{x \in D_e} f(x)$. Because both the considered point $x$ and $x_e^*$ belong to $D_e$, it means that they are not eliminated in the previous episode. Hence, we have

$$f(x_e^*) - f(x) \le \mu_{m_{e-1}}^*(x_e^*) + \beta_{(m_{e-1}+1)} \sigma_{m_{e-1}}(x_e^*) + \epsilon \sqrt{(2 + \lambda^{-1}) \gamma_{m_{e-1}}(k; D_{e-1})}$$
$$- \left( \mu_{m_{e-1}}^*(x) - \beta_{(m_{e-1}+1)} \sigma_{m_{e-1}}(x) - \epsilon \sqrt{(2 + \lambda^{-1}) \gamma_{m_{e-1}}(k; D_{e-1})} \right) \tag{66}$$

$$= \mu_{m_{e-1}}^*(x_e^*) - \beta_{(m_{e-1}+1)} \sigma_{m_{e-1}}(x_e^*) + 2\beta_{(m_{e-1}+1)} \sigma_{m_{e-1}}(x_e^*) - \mu_{m_{e-1}}^*(x)$$
$$- \beta_{(m_{e-1}+1)} \sigma_{m_{e-1}}(x) + 2\beta_{(m_{e-1}+1)} \sigma_{m_{e-1}}(x) + 2\epsilon \sqrt{(2 + \lambda^{-1}) \gamma_{m_{e-1}}(k; D_{e-1})}$$

$$\le \max_{x \in D_{e-1}} \{\mu_{m_{e-1}}^*(x) - \beta_{(m_{e-1}+1)} \sigma_{m_{e-1}}(x)\} + 2\beta_{(m_{e-1}+1)} \sigma_{m_{e-1}}(x_e^*) - \mu_{m_{e-1}}^*(x)$$
$$- \beta_{(m_{e-1}+1)} \sigma_{m_{e-1}}(x) + 2\beta_{(m_{e-1}+1)} \sigma_{m_{e-1}}(x) + 2\epsilon \sqrt{(2 + \lambda^{-1}) \gamma_{m_{e-1}}(k; D_{e-1})}$$

$$\le 2\beta_{(m_{e-1}+1)} \sigma_{m_{e-1}}(x_e^*) + 2\epsilon \sqrt{(2 + \lambda^{-1}) \gamma_{m_{e-1}}(k; D_{e-1})} + 2\beta_{(m_{e-1}+1)} \sigma_{m_{e-1}}(x) \tag{67}$$

$$\le 2\epsilon \sqrt{(2 + \lambda^{-1}) \gamma_{m_{e-1}}(k; D_{e-1})} + 4\beta_{(m_{e-1}+1)} \max_{x \in D_{e-1}} \sigma_{m_{e-1}}(x) \tag{68}$$

$$\le 2\epsilon \sqrt{(2 + \lambda^{-1}) \gamma_{m_{e-1}}(k; D_{e-1})} + 4\beta_{(m_{e-1}+1)} \sqrt{\frac{(1 + 2\lambda) \gamma_{m_{e-1}}(k; D_{e-1})}{m_{e-1}}}. \tag{69}$$

Here, Eq. (66) follows from applying Eq. (61) twice and using $m = m_{e-1}$. Next, Eq. (67) follows from the rule in Eq. (20) for retaining points in the algorithm and by noting that $x \in D_e$. Finally, Eq. (69) follows from Eq. (60).

We proceed to upper bound the total regret of our algorithm. We upper bound $R_T^*$ by providing an upper bound for $R_T$ and using the fact that $R_T^* \leq R_T + 2\epsilon T$. We also use $R_e$ to denote the regret incurred in episode $e$, $x_t^{(e)}$ to denote the selected point at time $t$ in episode $e$, and $E \leq \lceil \log_2 T \rceil$ to denote the total number of episodes. Finally, we consider $f \in \arg\min_{f \in \mathcal{F}_k(\mathcal{D};B)} \|f - f^*\|_\infty$ and denote $x^* = \arg\max_{x \in \mathcal{D}} f(x)$. It follows that

$$R_T \leq \sum_{e=1}^{E} R_e \tag{70}$$

$$\leq m_1 B + \sum_{e=2}^{E} \sum_{t=1}^{m_e} \left( f(x^*) - f(x_t^{(e)}) \right) \tag{71}$$

$$= B + \sum_{e=2}^{E} \sum_{t=1}^{m_e} \left( f(x^*) - f(x_e^*) + f(x_e^*) - f(x_t^{(e)}) \right) \tag{72}$$

$$\leq B + \sum_{e=2}^{E} m_e \left( 2\epsilon \sqrt{(2 + \lambda^{-1}) \gamma_{m_{e-1}}(\mathcal{D}_{e-1})} + 4\beta_{(m_{e-1}+1)} \sqrt{\frac{(1+2\lambda)\gamma_{m_{e-1}}(\mathcal{D}_{e-1})}{m_{e-1}}} \right)$$
$$+ \sum_{e=2}^{E} m_e (f(x^*) - f(x_e^*)) \tag{73}$$

$$\leq B + 2\epsilon \sqrt{(2 + \lambda^{-1}) \gamma_T(\mathcal{D})} \sum_{e=2}^{E} m_e + 4\beta_T \sqrt{(1 + 2\lambda)\gamma_T(\mathcal{D})} \sum_{e=2}^{E} m_e \sqrt{\frac{1}{m_{e-1}}}$$
$$+ \sum_{e=2}^{E} m_e (f(x^*) - f(x_e^*)) \tag{74}$$

$$\leq B + 6\epsilon T \sqrt{(2 + \lambda^{-1}) \gamma_T(\mathcal{D})} + 32\beta_T \sqrt{(1 + 2\lambda) T \gamma_T(\mathcal{D})}$$
$$+ \sum_{e=2}^{E} m_e (f(x^*) - f(x_e^*)) \tag{75}$$

In Eq. (71), we used $m_1 = 1$ and the fact that bounds on the RKHS norm imply bounds on the maximal function value if the kernel $k(\cdot, \cdot)$ is bounded (in our case, $k(x, x) \leq 1$ for every $x$):

$$|f(x)| = |\langle f, k(x, \cdot)\rangle_k| \leq \|f\|_k \|k(x, \cdot)\|_k = \|f\|_k \langle k(x, \cdot), k(x, \cdot)\rangle_k^{1/2} = B \cdot k(x,x)^{1/2} \leq B. \tag{76}$$

Finally, Eq. (73) follows from Eq. (69), and in Eq. (74) we used that $\beta_t$ is non-decreasing in $t$, and $\gamma_{m_{e-1}}(\mathcal{D}_{e-1}) \leq \gamma_T(\mathcal{D})$. To obtain Eq. (75), we used that $m_e = 2m_{e-1}$.

It remains to upper bound the term that corresponds to misspecified elimination, i.e., $\sum_{e=2}^{E} m_e(f(x^*) - f(x_e^*))$. First, we note that by Eq. (65) and monotonicity of $\gamma_t(\mathcal{D})$ both in $t$ and $\mathcal{D}$, we have

$$f(x^*) - f(x_e^*) = f(x^*) - f(x_2^*) + f(x_2^*) - \cdots - f(x_{e-1}^*) + f(x_{e-1}^*) - f(x_e^*) \tag{77}$$

$$< \sum_{i=1}^{e-1} 2\epsilon \sqrt{(2 + \lambda^{-1}) \gamma_{m_i}(\mathcal{D})} \tag{78}$$

$$\leq \sum_{i=2}^{e} 2\epsilon \sqrt{(2 + \lambda^{-1}) \gamma_{m_e}(\mathcal{D})} = 2(e-1)\epsilon \sqrt{(2 + \lambda^{-1}) \gamma_{m_e}(\mathcal{D})}. \tag{79}$$

Hence, we obtain

$$\sum_{e=2}^{E} m_e (f(x^*) - f(x_e^*)) \leq 2\epsilon \sqrt{(2 + \lambda^{-1}) \gamma_{m_E}(\mathcal{D})} \sum_{e=2}^{E} m_e(e-1) \tag{80}$$

$$\leq 6\epsilon T (\log T) \sqrt{(2 + \lambda^{-1}) \gamma_T(\mathcal{D})} \tag{81}$$

Finally, by combining Eq. (75) with Eq. (81), we obtain

$$R_T = O\left(\epsilon T(\log T)\sqrt{\gamma_T(\mathcal{D})} + \beta_T\sqrt{T\gamma_T(\mathcal{D})}\right). \tag{82}$$

The final result follows by upper-bounding $\beta_T$ as in Eq. (30). □

# E  Contextual misspecified setting results

To show the main result of section Section 4 (i.e., Proposition 1), we make use of the following theorem established in [32]. We recall it here for the sake of completeness.

**Theorem 3** (Theorem 5.5 in [32]). *Let the regret bounds for all base learners $i \in [M]$ be of the form:*

$$\mathcal{R}_i(t) = \min\{c_1\sqrt{t} + c_2\hat{\epsilon}_i t, \ t\},$$

*where $\hat{\epsilon}_i \in (0,1]$ and $c_1, c_2 > 1$ are quantities that do not depend on $\hat{\epsilon}_i$ and $t$. The regret of Algorithm 3 is bounded for all $T$ with probability at least $1 - \delta$:*

$$R_T^* = O\left(Mc_1\sqrt{T}\sqrt{\ln\tfrac{M\ln T}{\delta}} + Mc_2\hat{\epsilon}_{i^*}T\sqrt{\ln\tfrac{M\ln T}{\delta}} + Mc_1^2\right). \tag{83}$$

*Here, $i^*$ is any consistent base algorithm, i.e., $\hat{\epsilon}_i \le \hat{\epsilon}_{i^*}$ for all inconsistent algorithms.*

To apply this theorem together with our EC-GP-UCB algorithm, we note that the bound obtained for EC-GP-UCB in Theorem 1 is of the following form:

$$\tilde{\mathcal{R}}_i(t) = \min\{c_1'(B\sqrt{\gamma_t} + \gamma_t)\sqrt{t} + c_2'\epsilon_i\sqrt{\gamma_t}t, \ t\}. \tag{84}$$

In Algorithm 3, we make use of the following candidate upper bound instead:

$$\mathcal{R}_i(N_i(t)) = \min\{c_1'(B\sqrt{\gamma_T} + \gamma_T)\sqrt{N_i(t)} + c_2'\epsilon_i\sqrt{\gamma_T}N_i(t), \ t\}, \tag{85}$$

where $N_i(t)$ is the number of times algorithm $i$ was played up to time $t$. Here, due to monotonicity and $t \le T$, we have that $\gamma_T \ge \gamma_{N_i(t)}$, and hence we can replace $\gamma_{N_i(t)}$ with $\gamma_T$ in $\tilde{\mathcal{R}}_i(N_i(t))$. Moreover, we set $c_1 = c_1'(B\sqrt{\gamma_T} + \gamma_T)$ and $c_2'\sqrt{\gamma_T}$ in Theorem 3. Both $c_1$ and $c_2$ are then independent of $N_i(t)$. We also note that we can suitably set $\lambda$ (in Eq. (39)) such that $c_1, c_2 > 1$.

Hence, we can use the result of Theorem 3 to obtain the following regret bound:

$$R_T^* = O\left(Mc_1'(B\sqrt{\gamma_T} + \gamma_T)\sqrt{T} + c_2'\sqrt{\gamma_T}\hat{\epsilon}_{i^*}T)\sqrt{\ln\tfrac{M\ln T}{\delta}} + M(c_1')^2(B\sqrt{\gamma_T} + \gamma_T)^2\right). \tag{86}$$

We can then set $M = \lceil 1 + \log_2(\sqrt{T}/\gamma_T)\rceil$ and $\hat{\epsilon}_i = \frac{2^{1-i}}{\sqrt{\gamma_T}}$, because we have for $\hat{\epsilon}_1 = \frac{1}{\sqrt{\gamma_T}}$ that $\mathcal{R}_i(t) = t$ (which always holds) and, for $\hat{\epsilon}_M$ we have $\mathcal{R}_M(t) \le 2c_1'(B\sqrt{\gamma_T} + \gamma_T)\sqrt{T}$ which is only a constant factor away from the bound when $\epsilon = 0$. Hence, by taking a union bound over events that consistent and inconsistent candidate regret bounds hold and do not hold, respectively, we can then state that with probability at least $1 - M\delta$, it holds that

$$R_T^* = \tilde{O}\left((B\sqrt{\gamma_T} + \gamma_T)\sqrt{T} + \epsilon T\sqrt{\gamma_T} + (B\sqrt{\gamma_T} + \gamma_T)^2\right). \tag{87}$$