# OpenReview forum: "Misspecified Gaussian Process Bandit Optimization"
_NeurIPS.cc/2021/Conference — NeurIPS 2021 Poster_

### Official Review · Reviewer_qfKr · 2021-07-14

**Rating:** 6
**Confidence:** 3

**Summary:**

This paper considers GP-based global optimization in which the ground-truth function may be misspecified. The authors consider a hypothesis class defined by an RKHS ball of radius $B > 0$ and define misspecification as the ground-truth function not belonging to this class. The authors assume that the ground-truth function can still be approximated by a function from the hypothesis class in terms of the supremum norm: they assume that the distance (as defined by the supremum norm) between the ground-truth function and its best approximation from the hypothesis class is less than $\epsilon > 0$. Under this assumption, the authors derive regret bounds of the proposed approaches.

**Main Review:**

[Originality]
As the authors explain, the proposed approaches seem to be inspired by existing approaches proposed for different settings (e.g., linear bandits), but I feel that the originality is reasonable in that the authors adapt these previous approaches to the GP-based optimization setting.



[Clarity]
- The paper is well written, as details are explained to a reasonable extent and relevant papers are discussed well.

- But the authors miss one relevant result given in the following recent paper:

Wynne, G., Briol, F. X., & Girolami, M. (2021). Convergence guarantees for Gaussian process means with misspecified likelihoods and smoothness. Journal of Machine Learning Research, 22(123), 1-40.

Theorem 11 in Section 5.2 of the paper essentially implies a sublinear regret bound for a Bayesian optimization algorithm in a misspecified setting where the ground-truth function does not belong to the assumed RKHS.

- In line 156, the authors introduce observations $y_1, \dots, y_n$, and define the posterior mean function in Eq.(8). It is not clear how these observations are different from the observations $y_1^*, \dots, y_n^*$ defining the posterior mean function in Eq.(11). Please elaborate more. This is relevant to Lemma 2, which derives the difference between these two posterior mean functions.

- In Algorithm 2, what is the value of $\beta_{m_e+1}$? Is it given by  Eq.(13)?


[Quality]
- In lines 221-222, the authors claim that the $\epsilon T$ dependence in regret bounds is unavoidable for any algorithm in the misspecified kernelized setting, saying that this is shown in Appendix C. I think that this claim is misleading for the following reason:

-  In Appendix C the authors consider a ground-truth function whose function values are zero almost everywhere except on one input location. Such a ground-truth function is not very realistic. The authors should consider a more realistic misspecification setting; e.g., when the ground-truth function is rougher than the functions in the assumed RKHS but is still smooth. This is essentially the setting of Wynne et al., (2021), in which the assumed RKHS is a Sobolev space and the ground truth belongs to another Sobolev space of lesser smoothness.

- From the above reasoning, I would say that  $\epsilon T$ is too pessimistic, and the obtained regret bounds are not very informative (as they are not sublinear).

- Apart from the above theoretical issues, the paper lacks experimental assessment, and it is not very clear how the proposed approaches can perform well in practice. Since the authors propose new algorithms, I think there should be experiments to investigate their validity.

- Another issue arises from the fact that the value of $B$ implicitly determines $\epsilon$. If the ground-truth is not in the RKHS, to achieve the accuracy of $\epsilon$ in terms of the uniform norm, you need to make $B$ large enough. This means that there is a tradeoff between $B$ and $\epsilon$, and the performance of the proposed algorithms should be influenced by the specification of $B$. This point should also be discussed in my opinion.



[Significance]
As discussed for the [quality] criterion, I feel that the significance of this paper is limited, since

1) the assumed way of misspecification is too pessimistic (no assumption about the ground truth) and thus the regret bounds are not informative due to the $\epsilon T$ term;

2) no experiment is performed, so it's not clear whether the proposed algorithms can work well in practice;

and moreover

3) the stochastic contextual setting in Section 4, which assumes that the candidate actions are i.i.d.~generated from a probability distribution on the action space for each iteration, may not be realistic. The authors do not provide examples of applications where this setting appears. So I'm not sure whether the analysis in this setting is meaningful.


**Time Spent Reviewing:**

12 hours

---

> ### Author Response · Authors · 2021-08-09
> **Authors' Response to Reviewer qfKr**
>
> We thank the reviewer for the review and insightful questions. Responses to the questions are below (reference numbers correspond to the ones in the paper):
>
> -- We appreciate pointing out the related recent reference Wayne et al., 2021 that we’ll include in our paper. As noted by the reviewer, the setting considered in the referenced paper is different from ours and allows for a sublinear regret. In the setting that we consider, attaining sublinear regret depends on the joint dependency of $\epsilon$ and $T$, while for constant $\epsilon$ attaining sublinear regret is not possible.
>
> -- In Line 156, the observations $y_1, \dots, y_n$ correspond to some general realizable case, i.e., when the model is well-specified. We state this in the first sentence of Section 3.1. On the other hand, $y_1^*, \dots, y_n^*$ are noisy ground truth observations of $f^*$. We also define $\tilde{f}$ on Line 191, which is the “closest” realizable function in the given hypothesis class. In this context, we use $y_1, \dots, y_n$ to denote hypothetical noisy evaluations of the “closest” realizable function, i.e.,  in Eq. (7), $y_t$ then corresponds to $\tilde{f}(x_t) + \eta_t$.
>
> -- The reviewer is correct that $\beta_{m_e + 1}$ is set according to Eq. (13) by setting $t = m_e + 1$.
>
> -- (On the paragraph that includes: “The authors should consider a more realistic misspecification setting...”) We respectfully disagree with the reviewer. The work of Wynne et al., (2021) considers a relevant but a different problem setting. We believe that we should not consider the same problem as suggested by the reviewer, as it is already considered/solved in Wynne et al., (2021). Instead, we consider a different relevant misspecified problem studied in the bandit optimization and RL literature. The sublinearity of our regret bounds depends on the joint scaling of $\epsilon$ and $T$ and the goal is to obtain the optimal joint dependence. Moreover, obtaining upper bounds on regret in our setting is non-trivial, especially when $\epsilon$ is unknown, and such upper bounds cannot be achieved by standard Bayesian Optimization algorithms such as GP-UCB which makes this problem non-trivial. We also note that the $\epsilon T$ regret scaling is unavoidable in the simpler linear setting (see, e.g., [23]).  Finally, we would argue that our approach is more general as it makes no further structural assumptions when it comes to the ground truth function.
>
> -- (On the paragraph that includes: “you need to make B large enough”) We consider the case of fixed $B$ (i.e., which corresponds to a fixed misspecified hypothesis class). That is a common assumption in the kernelized bandit (i.e., the Frequentist setting) setting [9,34,38]. We note that the case mentioned by the reviewer is considered in, e.g., [4], where the universal squared-exponential kernel is used and the kernel lengthscales are decreased over time which increases $B$ (gradually allowing for more complex function classes) together with the sample complexity/regret rate.
>
> -- (On the stochastic contextual setting) We note that our contextual analysis in fact generalizes to the setting where $f(x, c)$ is defined on the joint action/context space, and contexts are assumed to be drawn i.i.d. (also a fixed set of actions). We chose to explain the less-general setting for the sake of simplifying the exposition. Addressing the generalization would require introducing: a composite kernel definition, new definitions of the regret and misspecification for $f^*(x,c)$, corresponding composite maximum information gain, and the contextual version of EC-GP-UCB akin to [21]. We will further elaborate on this and practical applications related to the stochastic contextual setting in the appendix. For example, user profiles are often considered as context in recommendation systems (Li et al., 2010), and i.i.d. might be a reasonable assumption for different users.
>
> *A Contextual-Bandit Approach to Personalized News Article Recommendation; L. Li, W. Chu, J. Langford, R. E. Schapire
>
> -- Similarly to many other misspecified bandit optimization and RL works [14,15,16,23,31,45] (including the works mentioned by the reviewers -- Wayne et al. 2021 and Camilleri et al. 2021) that provide no empirical evidence, the focus of our work is on theoretical results, and we believe that our theoretical findings should be the main factor in the final decision.

---

### Official Review · Reviewer_Ecau · 2021-07-16

**Rating:** 5
**Confidence:** 3

**Summary:**

The paper focuses on the kernelized bandit problem where the model can be misspecified: the objective function lies outside the given RKHS such that its distance (induced from the max-norm) to any member of the RKHS is at least $\epsilon$. It proposes 3 different algorithms which are capable of addressing the problem in different scenarios: known $\epsilon$, unknown $\epsilon$, and a contextual bandit setting. The performance of these methods are theoretically analyzed via the upper confidence bounds of the cumulative regrets which connect the algorithms’ performance to $\epsilon$, the time horizon, and the kernel.

**Limitations And Societal Impact:**

I agree with the authors that there is no significantly negative societal impact for this model.

**Main Review:**

Overall, the paper presents a complete treatment for the proposed problem: known $\epsilon$, unknown $\epsilon$, and a contextual bandit setting. I have some questions, especially on the theoretical analysis and the lack of empirical experiments in the paper.

1. (novelty) I am concerned about the novelty of the first contribution (lines 90-93) of the paper and the acknowledgement of [5] where the techniques in the proof and analysis (in section 3.2 and the appendix) are used. The analysis (the notion of the enlarged confidence bound in Lemma 1, its proof, the EC-GP-UCB algorithm, the proof of the cumulative regret bound, the explanation on the optimality of the dependence on $\epsilon T$ in the cumulative regret bound) follows closely to the analysis presented in [5]. While the paper describes the difference with [5] in lines 54-59, the authors may need to highlight the technical challenges/difference in the analysis in comparison to [5]. More importantly, while they are similar, the theoretical analyses (in section 3.2 and appendix) do not have any reference to [5]. Can the authors explain the reason [5] is not acknowledged/referred to when the analysis is presented?

2. (significance) Unlike the analysis for EC-GP-UCB which says that the dependence on $\epsilon T$ in the cumulative regret bound is unavoidable (line 221), there is not any analysis (or some justification/implications) about the dependence on $\epsilon T \sqrt{\gamma_T}$ for phased GP uncertainty sampling (in Theorem 2). In fact, the cumulative regret bound depends on $\epsilon T \log T \sqrt{\gamma_T}$ (in equation 85 in Appendix D where the polylog(T) factors are not hidden). It means that the average cumulative regret $R_T/T$ is unbounded when $T$ approaches infinity.

3. The contextual bandit setting will be more convincing if there are practical motivation examples for the setting.

4. The paper will be more complete if they have experimental results comparing the performance of EC-GP-UCB with other BO acquisition functions such as GP-UCB to highlight the benefit of modelling the misspecified error.


**Time Spent Reviewing:**

12

---

> ### Author Response · Authors · 2021-08-09
> **Authors' Response to Reviewer Ecau**
>
> We thank the reviewer for the insightful review. Responses to the questions are below (reference numbers correspond to the ones in the paper):
>
> -- One can think of our misspecified setting as the corrupted one in which at every round the corruption is bounded by $\epsilon$ leading to a total budget of $C = T \epsilon$. The UCB algorithm of [5] then incurs $T^{3/2}\epsilon \sqrt{\gamma_T}$ regret bound (due to corruptions) which is strictly suboptimal. The analysis of our EC-GP-UCB is simple and relies on the similar standard steps of expanding the mean estimator definition (as in [5]). These are also present and considered standard in various other kernelized bandits works, e.g., in [9]. The difference in the analysis comes from the treatment of misspecification constraint: In our setting, the misspecification at every round is the same and bounded by $\epsilon$, while in the corrupted one, the corruptions sum up over rounds (e.g., the adversary can also decide to exhaust its whole budget in the very first round). Hence, by treating these constraints differently in the analysis, we get the $\epsilon \sqrt{t}$ factor vs. $C$ dependence in the corrupted setting (in confidence bounds). We will elaborate on the difference w.r.t. [5] further (and include additional citations) in our paper as written above and suggested by the reviewer. Given these differences and other major contributions of our work, we hope that the reviewer will reconsider increasing the final score.
>
> -- (Significance) In line 221, we say that such dependence is unavoidable for any algorithm and not just EC-GP-UCB. Hence, both our algorithms are not “no-regret” (i.e., average regret does not converge to 0) in case of constant epsilon (attaining sublinear regret depends on the joint dependence of $\epsilon$ and $T$, while for constant $\epsilon$ attaining sublinear regret is not possible). The difference in the obtained regret bounds for the two proposed algorithms is in the multiplicative $\log(T)$ factor as noticed by the reviewer. This is a very minor detail given our main findings. This additional factor comes from the episodic nature of our second algorithm. We also note that for constant $\epsilon$, the average regret of other proposed linear and linear contextual bandit algorithms in the misspecified setting, see, e.g., Proposition 5.1 in [23], is also unbounded (in the mentioned limit).
>
> -- Similarly to many other misspecified bandit optimization and RL works [14,15,16,23,31,45] (including the works mentioned by the reviewers -- Wayne et al. 2021 and Camilleri et al. 2021) that provide no empirical evidence, the focus of our work is on theoretical results, and we believe that our theoretical findings should be the main factor in the final decision.

---

> > ### Comment · Reviewer_Ecau · 2021-08-17
> > **Response to Authors**
> >
> > Regarding the novelty of the paper (in comparison to [5]), I understand the difference in the problem definitions of this work and [5]. However, as I mentioned above, my concerns are: I do not know how this difference leads to additional challenges; and this work does not cite [5] in the theoretical analysis although it follows quite closely to [5].
> >
> > I am not mistaken, the main novelty in the analysis of EC-GP-UCB is the characterization of the corruption effect in Lemma 2 and equations 15,16 (through the enlarged confidence bound). But this idea was first introduced in [5] (the proof of Lemma 2 in Appendix D of this work is similar to Appendix A in [5] except for a minor change in bounding $||m_t||_2^2$). Furthermore, the following analysis (regret bound, the explanation on the optimality of the dependence on $\epsilon T$ in the cumulative regret bound) follows closely to that of [5]. Therefore, the fact that this work does not cite [5] in these analyses is not very satisfactory. Also, from the authors' response, I still do not know exactly the additional challenges in the analysis of EC-GP-UCB compared with [5].
> >
> > As the provided average regrets are unbounded, I still think the paper is more convincing with experimental results.
> >
> > My opinion on the paper remains the same after reading the authors' response.

---

> > > ### Author Response · Authors · 2021-08-17
> > > **Response to reviewer Ecau**
> > >
> > > We thank the reviewer for the response.
> > >
> > > We have explained that the result and algorithm of [5] lead to the strictly suboptimal regret guarantee when applied to our problem and hence cannot be directly invoked. We do not claim that our analysis introduces additional challenges. The analysis is simple (since this is a “warm-up” algorithm where we assume $\epsilon$ is known) and follows standard steps well-known in the GP bandit literature. Our previous sentence "by treating these constraints differently in the analysis, ..." refers to the change in terms of how $|| m_t ||_2^2$ is handled.
> > >
> > > Whether the average regret bounds are unbounded depends on the joint dependency of $T$ and $\epsilon$ (that is our main research focus). For example, in the case of the SE kernel, if $\epsilon = T^{-c}$, for some arbitrary small $c$, our algorithms are “no-regret”.

---

> > > > ### Comment · Reviewer_Ecau · 2021-08-31
> > > > **Response to Authors**
> > > >
> > > > Thank you for the clarification. However, I am still not totally convinced for the following reasons.
> > > >
> > > > Regarding the EC-GP-UCB, as I mentioned in the above response, the crucial part of the analysis is the characterization of the corruption effect via the enlarged confidence bound, which is first introduced in [5]. Therefore, saying that the analysis of this algorithm "follows well-known standard steps in the GP bandit literature" is not entirely correct. In other words, I expect the authors to give several existing BO works using the enlarged confidence bound and similar analyses to justify that the analysis of EC-GP-UCB is "well-known". In such a case, the fact that the paper do not cite [5] is more acceptable (even though I think [5] is still to be cited). On the other hand, if [5] is the only one (or very few) work(s) that introduces this technique in the BO literature, then the fact the analysis EC-GP-UCB utilizes the idea of [5] and does not cite [5] is inappropriate to me.
> > > >
> > > > Regarding the bounded cumulative regret when $\epsilon = T^{-c}$, this is not the problem setting that the authors describe in the paper (in line 117: "for some fixed $\epsilon > 0$"). Therefore, to say that the algorithm "no-regret" by modifying the problem setting is not convincing to me.
> > > >
> > > > While I acknowledge the contribution in developing Algorithms 2 and 3 in my main review (which explains my current score), it is also noted that point 3 of my review ("The contextual bandit setting (Algorithm 3) will be more convincing if there are practical motivation examples for the setting.") has not been answered.

---

> > > > > ### Author Response · Authors · 2021-09-01
> > > > > **Response to Reviewer**
> > > > >
> > > > > We would like to thank the reviewer for the detailed discussion and comments.
> > > > >
> > > > > We agree with the reviewer that our algorithm is also based on the enlarged confidence bounds as the one in [5]. We are unaware of other BO algorithms that make use of this idea (as also noted by the reviewer), but we also note that other misspecified (linear) bandit algorithms that we also draw inspiration from, use the idea of enlarged confidence bounds (and the upper confidence bound algorithms; e.g., [45], [23]). We will explicitly write this in the updated version and comment on the fact that the results/algorithm of [5], although very similar, lead to suboptimal regret guarantees for our problem. We will cite [5] in our proof and EC-GP-UCB section (main text) and state that this work is, up to our knowledge, the first to use this idea in the BO literature.
> > > > >
> > > > > The main steps of EC-GP-UCB analysis are given in Eq. 48-54. From these, Eq. 48-52 are "standard" as the exact steps are performed in the proof of standard confidence bounds (e.g., see [9] and Proof of Theorem 2 (Section C) and how the term $|k_t(x) (K_t + \lambda I)^{-1}) \varepsilon_{1:t}|$ is bounded is also used in our Eq. 48-52), while the last Eq. 54 simply follows from the problem setup.
> > > > >
> > > > > We wrote "fixed" to mean "specified as part of the problem" (i.e., not chosen by the algorithm) and not "fixed with respect to T", and we'll re-word this accordingly (no part of the analysis requires the latter).
> > > > >
> > > > > We thank the reviewer for acknowledging our main contributions. We also apologize for missing your point (3). We have included an answer in another response. On the stochastic contextual setting: We note that our contextual analysis in fact generalizes to the setting where $f(x,c)$ is defined on the joint action/context space, and contexts are assumed to be drawn i.i.d. (also a fixed set of actions). We chose to explain the less-general setting for the sake of simplifying the exposition. Addressing the generalization would require introducing: a composite kernel definition, new definitions of the regret and misspecification for $f^∗(x,c)$, corresponding composite maximum information gain, and the contextual version of EC-GP-UCB akin to [21]. We will further elaborate on this and practical applications related to the stochastic contextual setting in the appendix. For example, user profiles are often considered as a context in recommendation systems (Li et al., 2010), and i.i.d. might be a reasonable assumption for different users. Moreover, in the GP bandit literature, similar stochastic assumptions on the contexts are used, e.g., in [20, **].
> > > > >
> > > > > *A Contextual-Bandit Approach to Personalized News Article Recommendation; L. Li, W. Chu, J. Langford, R. E. Schapire
> > > > >
> > > > > ** Stochastic Bandits with Context Distributions; J. Kirschner, A. Krause

---

### Official Review · Reviewer_mPfu · 2021-07-17

**Rating:** 7
**Confidence:** 4

**Summary:**

This paper considers the problem of regret minimization for Gaussian process bandits. The authors focus on the setting where the model is misspecified and the true reward function is not contained in the underlying RKHS. They provide three algorithms: 1) a ucb based method for the setting where the misspecification error is known, 2) an elimination approach for when it is not known, 3) a contextual bandits version.

**Limitations And Societal Impact:**

Not addressed

**Main Review:**

Overall I enjoyed the paper and thought it was very well written. I do not have any serious issues with the claims made or the algorithms presented.

My main concern is comparing this work with a recent work:
https://arxiv.org/pdf/2105.05806.pdf.

Algorithm 2 is very similar to Algorithm 3 in the paper(RIPS for Regret Minimization). Indeed, greedily pulling the arm with the maximum variance should approximate a G-optimal design. This paper also basically carries out the program the authors describe on lines 248-254.

I am willing to give this paper an accept conditioned on a comparison to this previous work.

**Time Spent Reviewing:**

4

---

> ### Author Response · Authors · 2021-08-09
> **Authors' Response to Reviewer mPfu**
>
> We thank the reviewer for the positive review as well as the suggestion regarding the relevant work. We were unaware of the mentioned *contemporary* work Camilleri et al., 2021 that appeared on ArXiv roughly one week before the NeurIPS abstract deadline. Our response to the reviewer’s question regarding the main differences/similarities is below (reference numbers correspond to the ones in the paper):
>
> -- (Simplicity) Both our analysis and algorithm are *simpler* than the analysis and algorithm of Camilleri et al., 2021. We do not require *non-standard robust* estimators and simply use the standard (GP posterior/ kernelized ridge regression mean and variance estimators) ones that can be computed in the closed-form. On the other hand, Camilleri et al., 2021 require optimizing for a robust mean estimator. Hence, we show that a standard mean estimator is sufficient. On a conceptual level, their algorithm (and consequently, their analysis) is more complex as it requires robust mean estimation, optimizing to obtain distribution on the simplex, drawing samples, and solving a second optimization problem.
>
> -- (Practical aspects) Camilleri et al. 2021 focuses on finite sets only and do not specify a procedure for computing near-optimal design (i.e., consider their Eq. (6)) in case of an infinite action set. Our approach can handle infinite action sets in a similar way as the classical Bayesian optimization (e.g., like GP-UCB) algorithms do. It consists of a single acquisitions function (MAXVAR) that, in practice, can be maximized via standard global optimization solvers. Moreover, when it comes to practical computational benefits, we can further exploit recursive mean and variance updates as well as submodular set optimization techniques ([9,7]).
>
> -- (Complete problem treatment) As the reviewer noticed, Theorem 3 of Camilleri et al., 2021 contains the same regret scaling due to misspecification. However, we present a complete treatment of the misspecified problem: failure of the standard UCB approaches, impossibility result, known $\epsilon$ and UCB based algorithm, unknown $\epsilon$ and experimental design approach. Moreover, we also consider a contextual bandit setting. Most of these are not present in the referenced contemporary work, and we believe that our findings bring additional novel insights. While the previous algorithm extends the previous phased elimination of [3], our algorithm demonstrates the use of another different acquisition function together with standard and non-robust estimators.
>
> Finally, we commit to adding a paragraph that clearly explains the main differences and similarities with the referenced work in the camera-ready version.

---

### Decision · Program_Chairs · 2021-09-27

**Decision:**

Accept (Poster)

**Comment:**

Dear authors,

Following the discussion period, there is still some disagreement about this paper, hence I took a look at the paper.
After reading the paper, and taking into account the full discussion, I consider the paper is worth publication, assuming the authors incorporate all the clarification points in the final version (which I believe can be done with little effort).
Hence I recommend acceptation.